# A novel isoform of MAP4 organises the paraxial microtubule array required for muscle cell differentiation

Binyam Mogessie[1,2], Daniel Roth[1], Zainab Rahil[1†], Anne Straube[1]*

[1]Centre for Mechanochemical Cell Biology, Warwick Medical School, University of Warwick, Coventry, United Kingdom; [2]Cell Biology Division, MRC Laboratory of Molecular Biology, Cambridge, United Kingdom

**Abstract** The microtubule cytoskeleton is critical for muscle cell differentiation and undergoes reorganisation into an array of paraxial microtubules, which serves as template for contractile sarcomere formation. In this study, we identify a previously uncharacterised isoform of microtubule-associated protein MAP4, oMAP4, as a microtubule organising factor that is crucial for myogenesis. We show that oMAP4 is expressed upon muscle cell differentiation and is the only MAP4 isoform essential for normal progression of the myogenic differentiation programme. Depletion of oMAP4 impairs cell elongation and cell–cell fusion. Most notably, oMAP4 is required for paraxial microtubule organisation in muscle cells and prevents dynein- and kinesin-driven microtubule–microtubule sliding. Purified oMAP4 aligns dynamic microtubules into antiparallel bundles that withstand motor forces in vitro. We propose a model in which the cooperation of dynein-mediated microtubule transport and oMAP4-mediated zippering of microtubules drives formation of a paraxial microtubule array that provides critical support for the polarisation and elongation of myotubes.

*For correspondence: anne@ mechanochemistry.org

Present address: †Department of Chemical and Biomolecular Engineering, University of Illinois, Champaign, United States

Competing interests: The authors declare that no competing interests exist.

## Introduction

Skeletal muscle fibre formation requires a coordinated programme of morphological and biochemical changes in the differentiating cells. Upon differentiation, mono-nucleated myoblasts withdraw from the cell cycle and fuse to form syncytial myotubes (*Wakelam, 1985*). The microtubule cytoskeleton is required for these processes (*Bischoff and Holtzer, 1968*; *Holtzer et al., 1975*; *Toyama et al., 1982*) and undergoes reorganisation into an array of paraxial microtubules (*Warren, 1974*), which serves as template for contractile sarcomere formation (*Antin et al., 1981*; *Pizon et al., 2005*).

During myogenesis, microtubules are rearranged from a dynamic radial array to a parallel array of stable posttranslationally modified microtubules within the elongating cell (*Warren, 1974*; *Tassin et al., 1985*; *Saitoh et al., 1988*; *Gundersen et al., 1989*). Preventing detyrosination of tubulin or interfering with microtubule stabilisation by depletion of EB1 or MURF impairs expression of myogenic markers (*Spencer et al., 2000*; *Chang et al., 2002*; *Zhang et al., 2009*), thus suggesting that signalling through modified microtubules might control myogenic differentiation. On the other hand, changes in the regulation of microtubule dynamics at the cell cortex that neither affect the content of tubulin modifications nor the expression of differentiation markers can result in cell polarisation and fusion defects as caused by the depletion of EB3 (*Straube and Merdes, 2007*). This suggests a dual function of microtubules during the early stages of muscle cell differentiation to control (1) morphological changes and (2) biochemical composition, before later serving as structural templates for myofibrillogenesis.

A number of microtubule-associated proteins (MAPs) have been implicated in the organisation of microtubules into bundles. MAP2 and tau determine the spacing between microtubules in dendrites

**eLife digest** Skeletal muscles—which enable animals to move—are made up of large elongated muscle cells that span the entire length of the muscle. These cells contain stacks of structures called sarcomeres that enable the cells to contract and generate the force required for movement.

Cells called myoblasts elongate and fuse together at their tips to make the muscle cells. Within the myoblasts, long filaments called microtubules are arranged in an overlapping linear pattern. The filaments act as a template that helps the sarcomeres to align as the muscle cells form. A family of microtubule-associated proteins (or 'MAPs' for short) bind to microtubules and assist in organising the filaments, but it is not clear how they work.

Mogessie et al. used microscopy to observe the formation of the microtubule filaments in living myoblasts. The experiments show that the filaments progressively become more ordered as the myoblasts develop into muscle cells. Mogessie et al. identified a new member of the MAP family that is produced in myoblasts as soon as they start to form muscle fibres, and named it oMAP4. The microtubules in cells that make smaller amounts of this protein were more disorganised, and these cells were unable to fuse with each other to form muscle cells.

The experiments also found that oMAP4 can create links between different microtubules and act as a brake to prevent the filaments being moved excessively by motor proteins. Therefore, Mogessie et al. suggest that oMAP4 contributes to the formation of a strong and stable arrangement of filaments. This, in turn, allows the muscle cells to become very long.

Making more oMAP4 alone is not sufficient to form the elongated muscle cells. Therefore, the next challenge is to understand how other processes—such as the selective stabilisation of some microtubules and the movement of cell materials along the microtubules—cooperate to control muscle fibre formation.

and axons, but do not control the directionality of microtubules within those bundles (*Chen et al., 1992*). Proteins of the PRC1/MAP65/Ase1 family preferentially bundle microtubules in antiparallel orientation and are responsible for the stabilisation of the antiparallel microtubule overlaps in the spindle midzone during mitosis (*Loiodice et al., 2005*; *Gaillard et al., 2008*; *Subramanian et al., 2010*). Microtubule–microtubule sliding by Eg5, kinesin-1, kinesin-14, and dynein has been implicated in microtubule organisation and force generation (*Kapitein et al., 2005*; *Fink and Steinberg, 2006*; *Straube et al., 2006*; *Braun et al., 2009*; *Fink et al., 2009*; *Lu et al., 2013*; *Tanenbaum et al., 2013*), and we begin to understand how the interplay of motors and MAPs organises particular microtubule arrangements (*Janson et al., 2007*; *Bieling et al., 2010*; *Braun et al., 2011*).

Here, we show that the microtubule array in myoblasts is highly motile and becomes increasingly parallelised and immobilised as cells progress through the differentiation programme. We identify a previously uncharacterised differentially regulated isoform of microtubule-associated protein MAP4, called oMAP4, as a key organiser of microtubules in differentiating cells. Depletion of oMAP4 results in microtubule misalignment and increased microtubule motility in differentiating muscle cells. This results in defects in myogenic progression, cell polarisation, and cell–cell fusion. We further show that oMAP4 zippers preferentially antiparallel microtubules in vitro and restricts motor-driven microtubule sliding in differentiating muscle cells. Based on our own data, we propose a model whereby the cooperation of motor-driven microtubule–microtubule sliding and oMAP4-mediated zippering organises the microtubule cytoskeleton in differentiating muscle cells to support and govern cell polarisation and differentiation.

## Results

### Microtubules become progressively more ordered and immobile during muscle differentiation

To characterise microtubule organisation in differentiating muscle cells, we determined filament orientation, motility, and growth characteristics of microtubules at different stages during differentiation of C2C12 cells. As reported previously, we found that microtubule organisation changes from a radial, centrosome-dominated array in undifferentiated cells, to a paraxial array in

myotubes (*Figure 1A*) (*Warren, 1974*; *Tassin et al., 1985*). Microtubule growth directionality relative to the longitudinal axis of the cell was determined by tracking EB3-tdTomato-labelled microtubule ends. The asymmetry in the distribution of microtubule growth angles increases significantly during the first two days of differentiation (*Figure 1B–D*, *Figure 1—figure supplement 1*). This suggests that guided growth of microtubules (probably along existing microtubules) contributes to the progressively more ordered parallel microtubule array in differentiating cells. Furthermore, microtubules within the array are highly motile in undifferentiated myoblasts as visualised by the photoactivation of paGFP-Tubulin or conversion of mEos2-Tubulin (*Figure 1E,F*, *Videos 1, 2*). In differentiating muscle cells, microtubules become very stable and static as seen by the low frequency and speed of microtubule-sliding movements and by diminished loss of microtubules from the photoactivated region due to depolymerisation (*Figure 1E–H*, *Figure 1—figure supplement 2*, *Video 3*). The reduction in microtubule movements could be due to the differential regulation of motors that drive microtubule sliding. Conventional kinesin as well as dynein has been implicated in microtubule–microtubule sliding and microtubule movement along the cell cortex in other cell systems (*Rusan et al., 2002*; *Fink and Steinberg, 2006*; *Straube et al., 2006*; *Bicek et al., 2009*; *Jolly et al., 2010*; *Samora et al., 2011*; *Lu et al., 2013*). Both motors contribute to microtubule movements in myoblasts as the frequency of microtubule movements was reduced more than threefold upon depletion of either dynein heavy chain or Kif5b (*Figure 1E–H*, *Figure 1—figure supplement 3*, *Videos 4, 5*). However, Kif5b and dynein were continuously expressed during muscle differentiation (*Figure 1I*), suggesting that other factors prevent motor-dependent microtubule sliding and promote parallel microtubule array formation during myogenesis.

## Three MAP4 isoforms are expressed in differentiating muscle cells

Structural MAPs are candidates for a role in organising parallel microtubule arrays and regulating motor activity. MAP4, the only non-neuronal member of the MAP2/Tau family, has been reported to stabilise and bundle microtubules (*Aizawa et al., 1991*; *West et al., 1991*; *Ookata et al., 1995*; *Nguyen et al., 1997*, *1998*; *Hasan et al., 2006*). Our previous finding that MAP4 can limit force generation by dynein motors (*Samora et al., 2011*) suggested that MAP4 might prevent dynein-driven microtubule motility in muscle cells. Mouse skeletal muscle cells express tissue-specific isoforms of MAP4 (*Mangan and Olmsted, 1996*). To investigate the differential regulation of experimentally confirmed and predicted MAP4 transcripts (*Figure 2A*), we performed RT-PCR analysis of total RNA isolated from differentiating C2C12 cells. We confirmed the continuous expression of the ubiquitous isoform uMAP4 and upregulation of muscle-specific mMAP4 24 hr post induction of differentiation (*Figure 2B*). Interestingly, we found that a previously uncharacterised isoform, with a unique 48 kD projection domain, was also upregulated after 24 hr of differentiation. In addition to muscle, this isoform is also highly expressed in brain tissue (*Figure 2—figure supplement 1*). We refer to this isoform as oMAP4. Further analysis revealed that uMAP4, mMAP4, and oMAP4 transcripts were each expressed as variants with three, four, or five tau-like microtubule binding repeats due to alternative splicing of exons 14 and 15 (*Figure 2—figure supplement 2*). To confirm that these transcripts encode proteins, we generated specific antibodies for mMAP4 and oMAP4 and probed whole cell lysates of C2C12 cells on Western blots. Both mMAP4 and oMAP4 protein levels increase between 24 and 48 hr after differentiation (*Figure 2C,D*). As expected, GFP-fusion proteins of uMAP4, mMAP4, and oMAP4 decorated microtubules along their length in C2C12 cells (*Figure 2—figure supplement 3*).

## oMAP4 is required for muscle cell elongation and fusion

To investigate their involvement in microtubule organisation in muscle cells, we depleted each MAP4 isoform using a vector-based RNA interference approach. GFP-Tubulin was co-expressed with the short hairpin RNAs (shRNAs) to serve as a marker to detect successful transfection. Efficient depletion was confirmed by Western blotting of FACS-sorted GFP-positive cells and by immunofluorescence (*Figure 2—figure supplements 4, 5*). Muscle differentiation phenotypes were assessed as (1) the ability of single nucleated cells to elongate and (2) the efficiency of cells to fuse and form syncytia containing three or more nuclei (*Straube and Merdes, 2007*). 48 hr after differentiation, the average length of myoblasts treated with control shRNA was 107 ± 40 μm. Depletion of uMAP4 did not significantly affect this (105 ± 36 μm, p = 0.4; *Figure 2E,F*). Depletion of mMAP4 allowed myoblasts to

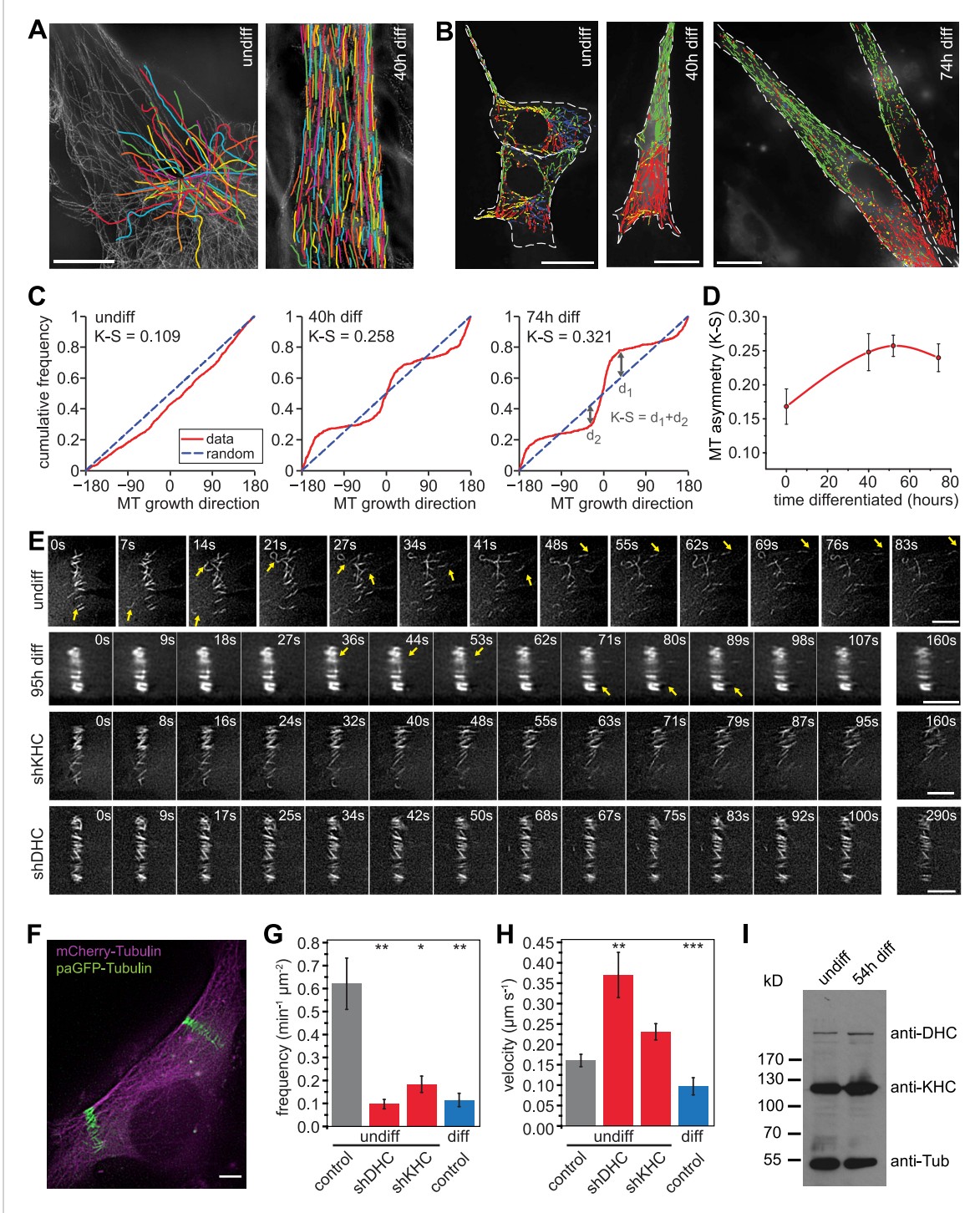

**Figure 1**. Microtubules are arranged in stable paraxial arrays during muscle cell differentiation. (**A**) Structured illumination microscopy of anti-tubulin-stained C2C12 cells pre/post induction of muscle differentiation as indicated. Microtubule filaments have been manually traced to highlight arrangement. Scale bar 10 μm. (**B**) Tracks of EB3-GFP in C2C12 cells at different stages of differentiation. Directionality is colour-coded (green and red: ±45° to longitudinal cell axis, blue and yellow: perpendicular to cell axis ±45°). Cell outlines are indicated with dashed white line. Scale bars 20 μm. (**C**) Cumulative distribution of MT growth angles for example cells shown in (**B**). Kuiper statistics (K–S) is calculated as a measure for microtubule alignment as the sum of the maximum deviations d1 and d2 from a random distribution. See *Figure 1—figure supplement 1* for angular histograms. (**D**) Average MT asymmetry of differentiating myoblasts. Data show mean ± SEM for 4–9 cells with >5000 microtubule tracks per condition. (**E** and **F**) Motility of paGFP-Tubulin-labelled microtubule segments in myoblasts pre/post induction of differentiation and after depletion of dynein (shDHC) and kinesin-1 (shKHC). Bar-shaped patterns were activated perpendicular to the microtubule orientation using mCherry-Tubulin as marker (**F**). An individual bar-shaped activation

*Figure 1. continued on next page*

*Figure 1. Continued*

pattern is shown in (**E**) for each condition. Arrows highlight microtubule-sliding events. Scale bars are 5 µm. See supplementary *Videos 1–5*. (**G** and **H**) Frequency and velocity of microtubule sliding events observed following photoactivation of tubulin segments. Data show mean ± SEM, n = 17–51 activated patterns. Asterisks indicate significant difference from undifferentiated control cells (*p < 0.05, **p < 0.005, ***p < 0.0005). (**I**) Immunoblotting of C2C12 cell extracts pre/post induction of differentiation for DHC and KHC. Tubulin serves as loading control.

The following figure supplements are available for figure 1:

**Figure supplement 1**. Microtubule growth orientation.

**Figure supplement 2**. Dissipation of photoconverted microtubule labelling.

**Figure supplement 3**. Verification of dynein and kinesin depletion.

elongate to an average length of 143 ± 58 µm, which is significantly longer than control cells. Myoblast fusion efficiency was not significantly affected following the depletion of uMAP4 or mMAP4 (*Figure 2G*). In contrast, the depletion of oMAP4 caused severe defects in myoblast elongation and fusion. oMAP4-depleted cells had an average length of 82 ± 39 µm, which is significantly shorter than control myoblasts (p << 0.001, *Figure 2E,F*). Furthermore, cell–cell fusion efficiency was reduced more than fourfold following depletion of oMAP4 (*Figure 2G*). Importantly, co-transfection of an RNAi-resistant version of oMAP4 (FLAG-oMAP4$^{RIP}$) rescued the myoblast elongation and fusion defects caused by oMAP4 shRNA (*Figure 2H,I*, *Figure 2—figure supplement 6*). The overexpression of uMAP4 was not able to rescue oMAP4-depletion phenotypes (*Figure 2I*). Our data collectively demonstrate that oMAP4 is specifically required for morphogenesis and cell–cell fusion during muscle cell differentiation.

## oMAP4 depletion leads to disorganised microtubules

To understand which processes in the myogenic programme depend on oMAP4, we analysed the timing of myogenic events. We find a severe delay and reduction in the expression of embryonic myosin, a structural protein required for sarcomere formation, in oMAP4-depleted cells (*Figure 3A–C*). Likewise, the relocation of centrosomal proteins to the nuclear surface of myoblasts, a characteristic event during myogenesis (*Tassin et al., 1985*; *Srsen et al., 2009*), is severely delayed in oMAP4-depleted cells (*Figure 3D,E*). These results confirm that progression through the myogenic differentiation programme depends on oMAP4. Previous work has shown that interference with microtubule stability and the acquisition of posttranslational modifications of tubulin result in similar myogenesis defects (*Spencer et al., 2000*; *Chang et al., 2002*; *Zhang et al., 2009*). Depletion of oMAP4 did not result in a reduction of tubulin acetylation (*Figure 3F*), perhaps because other microtubule stabilisers were still present. Nevertheless, the microtubule cytoskeleton appeared disorganised in oMAP4-depleted cells (*Figure 3A,B*). To quantitatively assess microtubule network organisation and distinguish this from effects due to different morphology and differentiation status, we selected mono-nucleated elongated myoblasts treated with control and oMAP4 shRNA and manually traced GFP-Tubulin-labelled microtubules in equivalent sections of these cells (*Figure 3G*,

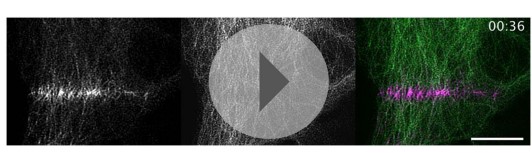

**Video 1.** Photoconversion of mEos2-Tubulin in an undifferentiated C2C12 myoblast showing converted (left panel and magenta in right panel) and non-converted channels (middle panel and green in right panel). Scale bar: 10 µm.

*Videos 6, 7*). We found that the microtubule network was largely parallel in control cells with ~80% of microtubules oriented ±15° of the longitudinal cell axis. This was reduced to ~50% in oMAP4-depleted cells (*Figure 3H*). Microtubule orientation was fully rescued by expression of an RNAi-resistant oMAP4 construct (*Figure 3—figure supplement 1*). Depletion of mMAP4 increased paraxial microtubule alignment and uMAP4 depletion slightly reduced it (*Figure 3—figure supplement 1*). We confirmed the microtubule alignment defect caused by

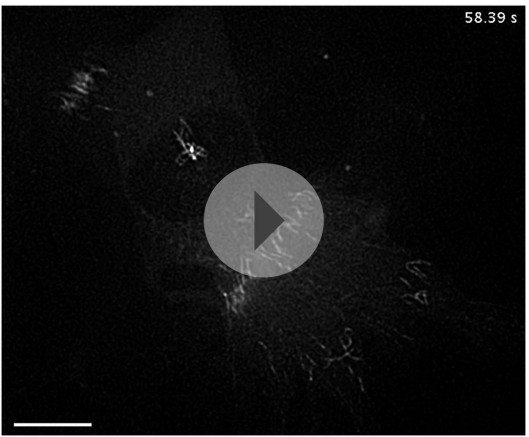

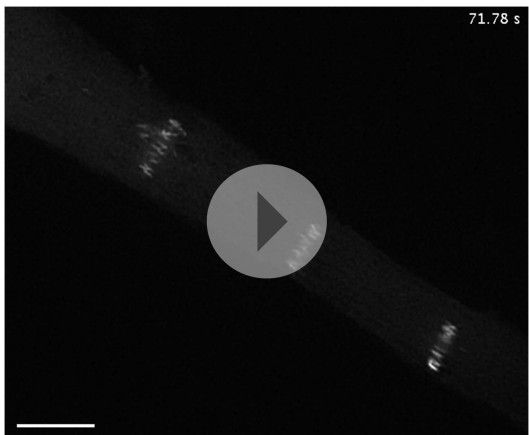

**Video 2.** Photoactivation of bar-shaped patterns of paGFP-Tubulin in an undifferentiated C2C12 myoblast. Scale bar: 10 μm.

**Video 3.** Photoactivation of bar-shaped patterns of paGFP-Tubulin in a 94-hr differentiated C2C12 myoblast. Scale bar: 10 μm.

oMAP4 depletion results by tracking growing microtubule ends labelled with EB3-tdTomato (*Figure 3I*, *Videos 8, 9*). While microtubule growth speed and duration were only slightly affected, the orientation of microtubule growth deviated more from the longitudinal cell axis when oMAP4 was depleted compared to control cells (*Figure 3J–L*, *Figure 3—figure supplement 2*).

## oMAP4 prevents motor-driven microtubule sliding in cells

We hypothesised that a failure in the guidance of microtubule assembly along existing filaments could be the underlying cause of the disorganised microtubule network. Alternatively, microtubule sliding and looping as observed in undifferentiated cells could disorganise the microtubule network in oMAP4-depleted cells. To distinguish between these possibilities, we analysed microtubule motility in differentiated, shRNA-treated cells, again selecting elongated, mono-nucleated cells to exclude effects due to morphology alone. As described above, microtubule motility was strongly suppressed in differentiating muscle cells (*Figures 1E–H, 4a*). Depletion of oMAP4 resulted in a more than

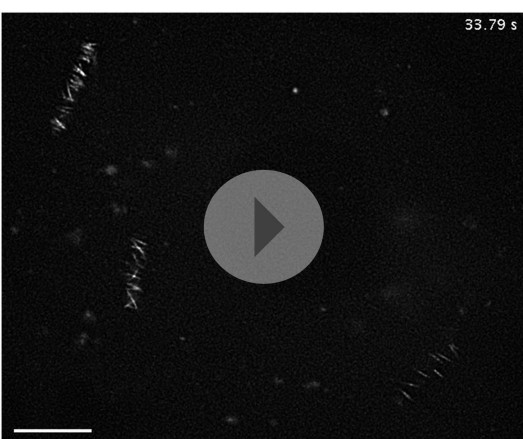

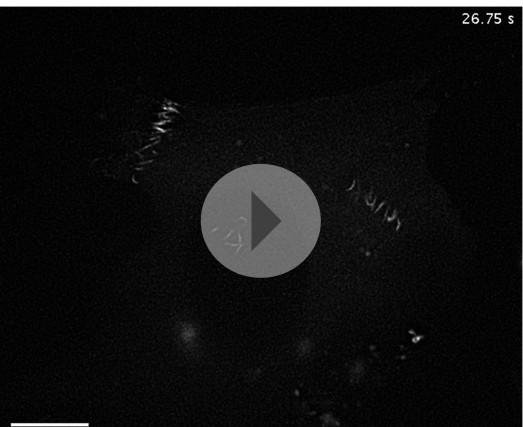

**Video 4.** Photoactivation of bar-shaped patterns of paGFP-Tubulin in an undifferentiated C2C12 myoblast treated with shRNA against dynein heavy chain. Scale bar: 10 μm.

**Video 5.** Photoactivation of bar-shaped patterns of paGFP-Tubulin in an undifferentiated C2C12 myoblast treated with shRNA against kinesin heavy chain (Kif5b). Scale bar: 10 μm.

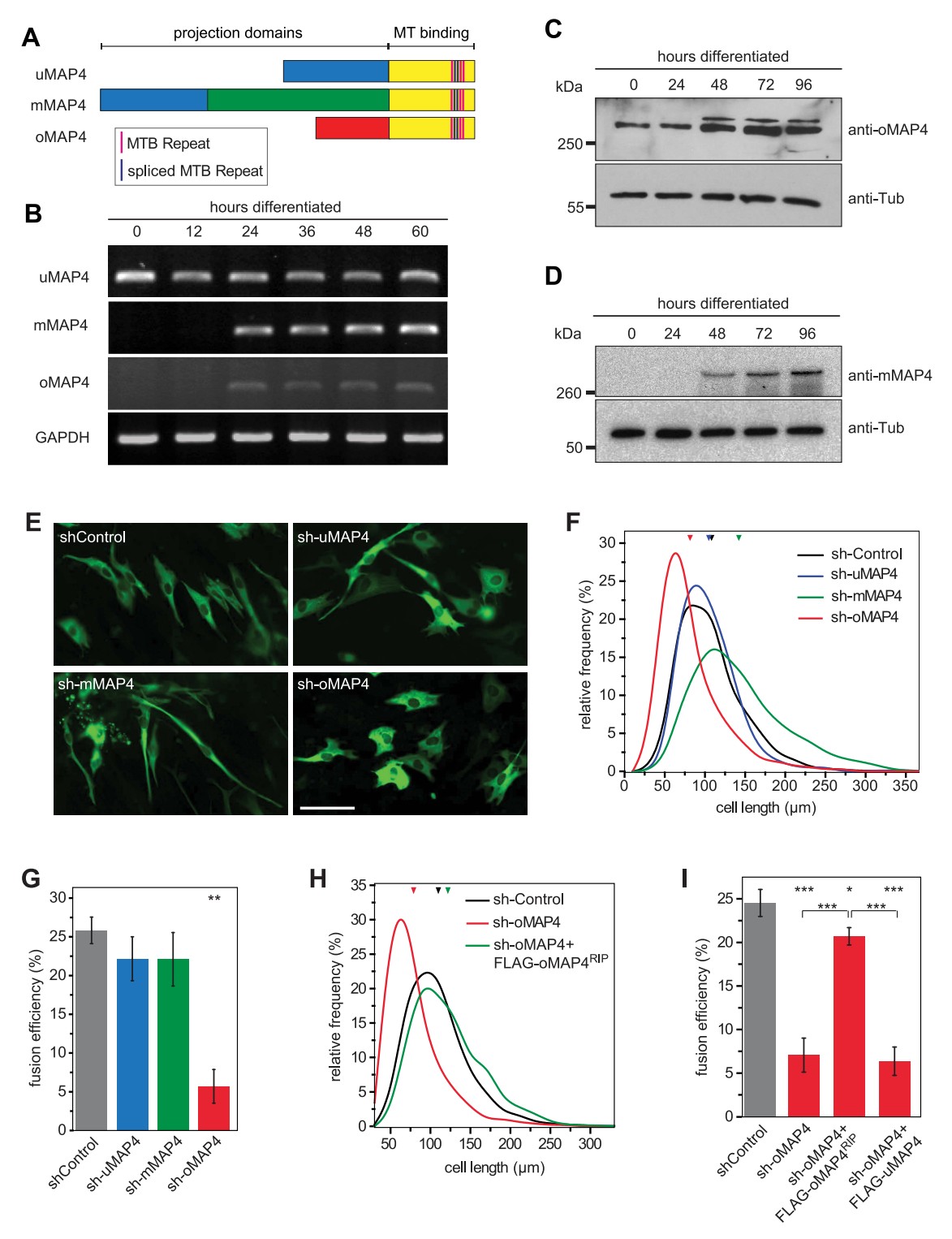

**Figure 2**. oMAP4 is required for myoblast elongation and fusion. (**A**) Domain organization of MAP4 isoforms. Green and red boxes represent isoform-specific regions in the projection domains of mMAP4 and oMAP4, respectively. Note that all three isoforms are expressed with 3, 4, or 5 tau-like MT binding repeats. (**B**) RT-PCR-based MAP4 expression analysis of RNA samples isolated from 0–60 hr differentiated C2C12 cells. Glyceraldehyde-3-phosphate dehydrogenase (GAPDH) was used as loading control. (**C** and **D**) Immunoblotting of extracts from 0 to 96 hr differentiated C2C12 cells using antibodies against the N-terminus of oMAP4 (**C**) or the muscle-specific insertion of mMAP4 (**D**). Tubulin was used as a loading control. (**E**) Examples of shRNA-treated cells co-expressing GFP-Tubulin 48 hr after induction of differentiation. Scale bar 50 μm. (**F**, **H**) Distribution of cell lengths after 48 hr

*Figure 2. continued on next page*

*Figure 2. Continued*

differentiation following treatment with indicated shRNAs. 1000–1900 cells were measured from three independent experiments for each condition. Small triangles represent mean values. (**G**, **I**) Myoblast fusion analyses of C2C12 cells treated with indicated shRNAs after 56 hr differentiation.1000–1200 cells were analysed for each condition and only cells with three or more nuclei were scored as fused. Data are collected from three experiments for each condition and are presented as mean ± S.E.M. Asterisks indicate significant difference from shControl treatment or between pairs of samples as indicated (*$p < 0.05$, **$p < 0.005$, ***$p < 0.0005$).

The following figure supplements are available for figure 2:

**Figure supplement 1**. Expression of major MAP4 isoforms.

**Figure supplement 2**. C2C12 cells express three major MAP4 isoforms with variable numbers of MT binding repeats.

**Figure supplement 3**. Microtubular localisation of eGFP-tagged MAP4 isoforms in C2C12 cells.

**Figure supplement 4**. Verification of mMAP4 depletion by immunofluorescence.

**Figure supplement 5**. Verification of MAP4 depletion by immunoblotting.

**Figure supplement 6**. Verification of oMAP4 depletion and RNAi-protected rescue construct by immunoblotting.

fourfold increase in the frequency of microtubule sliding events, and these were significantly faster than those in control cells (*Figure 4A–D*). In addition, the rare sliding events in differentiated control cells were usually restricted to movements parallel to the long axis of the cell, while microtubules were observed to loop in oMAP4-depleted cells, similarly to undifferentiated cells (*Figure 4E*, *Videos 1–3, 10, 11*). These observations suggest that oMAP4 acts to maintain the parallel microtubule network by restricting microtubule movement, especially those that are off-axis.

As we demonstrated in a previous study that uMAP4 limits force generation by dynein to prevent excessive movement of astral microtubules along the cell cortex (*Samora et al., 2011*), we wondered whether oMAP4 prevents microtubule sliding by inhibiting dynein function directly or whether oMAP4 crosslinks microtubules and acts as a brake to all motor-driven sliding. In order to distinguish between these possibilities, we co-depleted dynein from oMAP4-depleted cells. If oMAP4 would act exclusively through dynein, we posited this should rescue the depletion phenotypes. Depletion of dynein heavy chain alone did not significantly affect the already low-microtubule sliding frequencies in differentiated cells, although the velocity of microtubule sliding was reduced (*Figure 4B–D*). Cells co-depleted for dynein and oMAP4 still showed a high frequency of microtubule-sliding events (*Figure 4B,C*), although fast sliding events and looping were reduced (*Figure 4D,E*). Consequently, dynein co-depletion slightly alleviated the microtubule disorganisation caused by oMAP4 depletion, but did not fully rescue it (*Figure 4—figure supplement 1*). These results suggest that oMAP4 prevents microtubule sliding by crosslinking microtubules rather than by inhibiting dynein force generation per se. Indeed, expression of GFP-oMAP4 in undifferentiated myoblasts resulted in a significant increase in strong microtubule bundles, while expression of GFP-uMAP4 had no effect (*Figure 4F,G*).

## oMAP4 crosslinks antiparallel microtubules in vitro

To directly test the hypothesis that oMAP4 is a crosslinker, we recombinantly expressed and purified oMAP4 and GFP-oMAP4 from *Escherichia coli* (*Figure 5A*). Using in vitro microtubule co-sedimentation assays, we confirmed microtubule-binding activity of the purified proteins (*Figure 5B*). When Taxol- or GMP-CPP-stabilised microtubules were incubated with 60-nM oMAP4, we frequently observed microtubule bundles and structures with crossovers (*Figure 5C–F*). This confirmed that oMAP4 has microtubule cross-linking activity. We next asked whether oMAP4 has the ability to organise dynamic microtubules into antiparallel bundles in vitro. To do this, we used total internal reflection (TIRF) microscopy to visualise microtubules assembled from biotinylated microtubule seeds immobilised on streptavidin-coated coverslips. In control chambers, microtubules

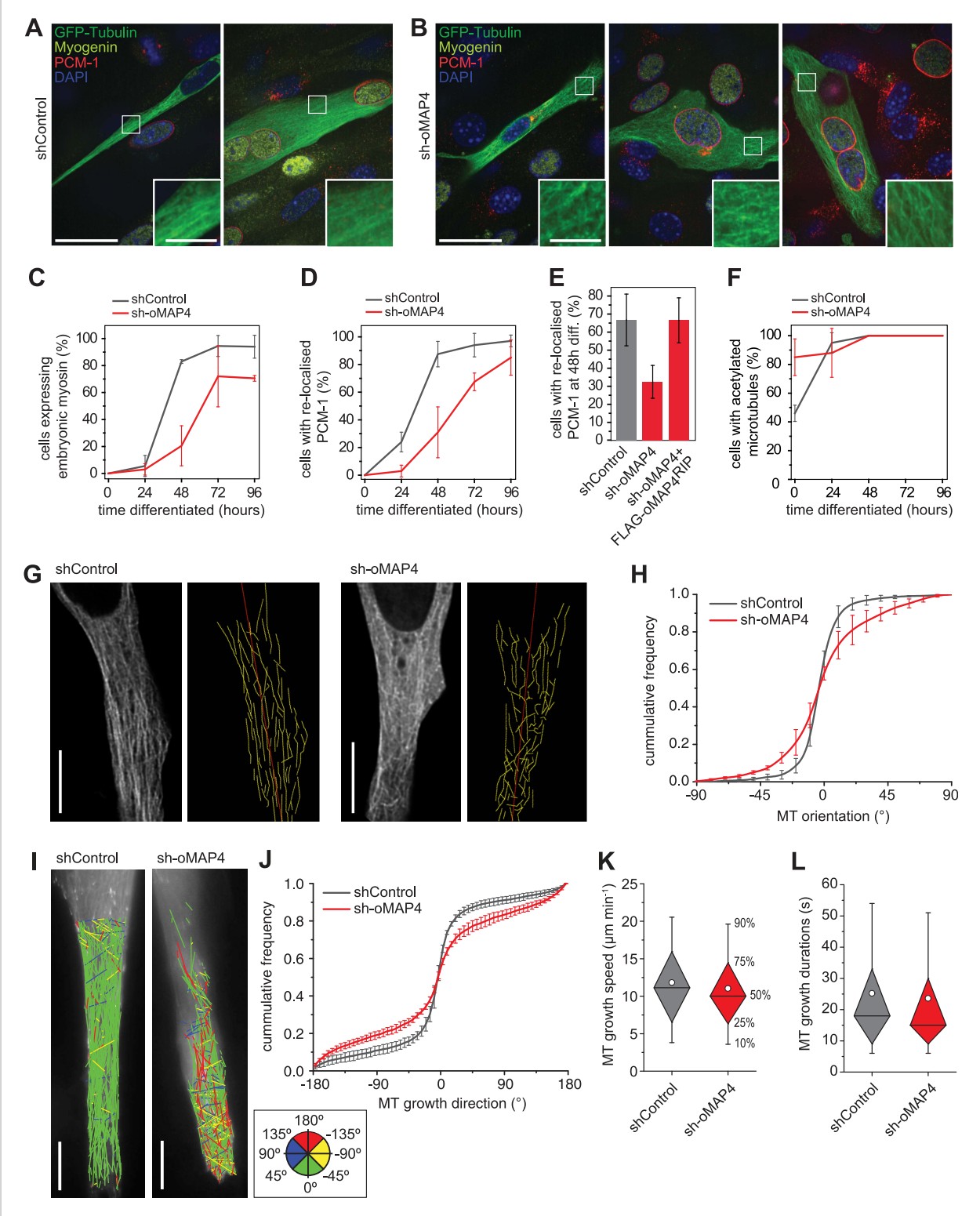

**Figure 3**. oMAP4 is required for the parallel arrangement of microtubules in differentiating muscle cells. (**A** and **B**) C2C12 myoblasts 48 hr after induction of differentiation treated with shRNA as indicated and stained for Myogenin (a marker for differentiating myoblasts, yellow), PCM-1 (red) and DAPI (blue). GFP-Tubulin (green) indicates successful transfection with shRNA. Insets show higher magnification of microtubule arrangement. Scale bars 25 μm, 5 μm in insets. (**C**) Timecourse of expression of embryonic myosin, a marker of myogenic differentiation. (**D**) Timecourse of relocalisation of PCM-1 from a focus around the centrosome to the surface of the nucleus. Cells with a complete nuclear ring were scored as positive. Data in **C**, **D** show mean ± SD, n = 30–50

*Figure 3. continued on next page*

*Figure 3. Continued*

cells from 2 experiments. (**E**) RNAi rescue experiment of delayed PCM-1 relocalisation phenotype. Data show mean ± SEM of 3 experiments with 50–60 cells each. (**F**) Timecourse of accumulation of cells with acetylated tubulin during muscle cell differentiation. Data in show mean ± SD, n = 30–50 cells from 2 experiments. (**G**) Manual tracing of microtubule filaments (yellow) relative to the longitudinal cell axis (red) in elongated mono-nucleated cells selected after 48 hr differentiation and shRNA treatment as indicated. Scale bars 10 µm. See supplementary *Videos 6, 7*. (**H**) Microtubule directionally from data as in g shown as cummulative frequency distribution. Data show mean ± SD, n = 3 experiments, 1522–1958 MTs. (**I**) Automatic tracking of EB3-tdTomato comets in elongated mono-nucleated cells selected after 48 hr differentiation and shRNA treatment as indicated. Direct lines from start to end of track are shown and colour-coded for direction relative to longitudinal axis of cell as indicated in legend. Scale bars 10 µm. See supplementary *Videos 8, 9*. (**J**) Distribution of microtubule growth angles obtained from data as in **H**. Data show mean ± SD, n = 3 experiments, 6251–6382 tracks. (**K** and **L**) Microtubule growth speed and duration was determined from EB tracks as in I. Pooled data shown as statistical box plots with percentiles as indicated.

The following figure supplements are available for figure 3:

**Figure supplement 1**. Microtubule orientation in depleted cells.

**Figure supplement 2**. Microtubule growth orientation in depleted cells.

continued growing without changing direction when they encountered other microtubules and microtubules only overlapped when they happened to grow in the same direction (*Figure 6A,B*, *Video 12*). The addition of GFP-oMAP4 promoted zippering of those growing microtubules that encountered each other at shallow angles (*Figure 6A–C*; *Video 13*). To assess whether oMAP4 was specific for the orientation of the microtubules, we determined microtubule polarity based on the growth characteristics of the microtubule ends observed in the video (*Figure 6C*) and determined the rate of microtubule zippering relative to the incident angle of the two microtubules. No microtubule-zippering events were observed at angles between 25° and 150°, suggesting that oMAP4 can only generate forces to bend microtubules by up to 30°. Furthermore, oMAP4 showed a strong preference for zippering antiparallel-oriented microtubules (*Figure 6B,C*).

Another known antiparallel microtubule-bundling protein, PRC1 is a dimer that accumulates specifically in antiparallel-microtubule overlaps in the spindle midzone (*Subramanian et al., 2010*). We observed that PRC1 was more potent to bundle-stabilised microtubules free in solution than oMAP4 (*Figure 5E,F*). However, PRC1 was not able to zipper microtubules in our assays using dynamic microtubules growing from immobilised seeds (*Figure 6B*). This is in agreement with the literature that described PRC1 to specifically bind to antiparallel-microtubule overlaps that form when microtubules 'occasionally encountered each other in a plus end-to-plus end configuration' (*Bieling et al., 2010*)

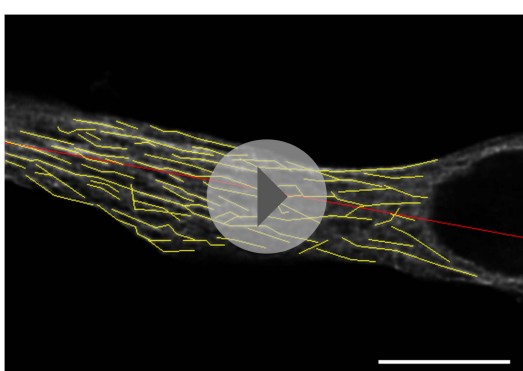

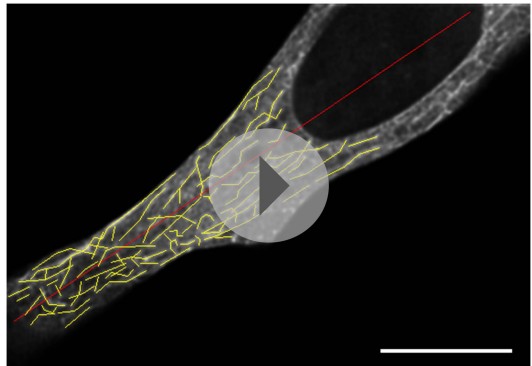

**Video 6.** Microtubule orientation in a 48 hr differentiated C2C12 myoblast treated with shControl co-expressing GFP-Tubulin. Manual tracing of microtubule cytoskeleton (yellow lines) and main cell axis (red line) shown as used for analysis. Scale bar: 10 µm.

**Video 7.** Microtubule orientation in a 48-hr differentiated C2C12 myoblast treated with sh-oMAP4 co-expressing GFP-Tubulin. Manual tracing of microtubule cytoskeleton (yellow lines) and main cell axis (red line) shown as used for analysis. Scale bar: 10 µm.

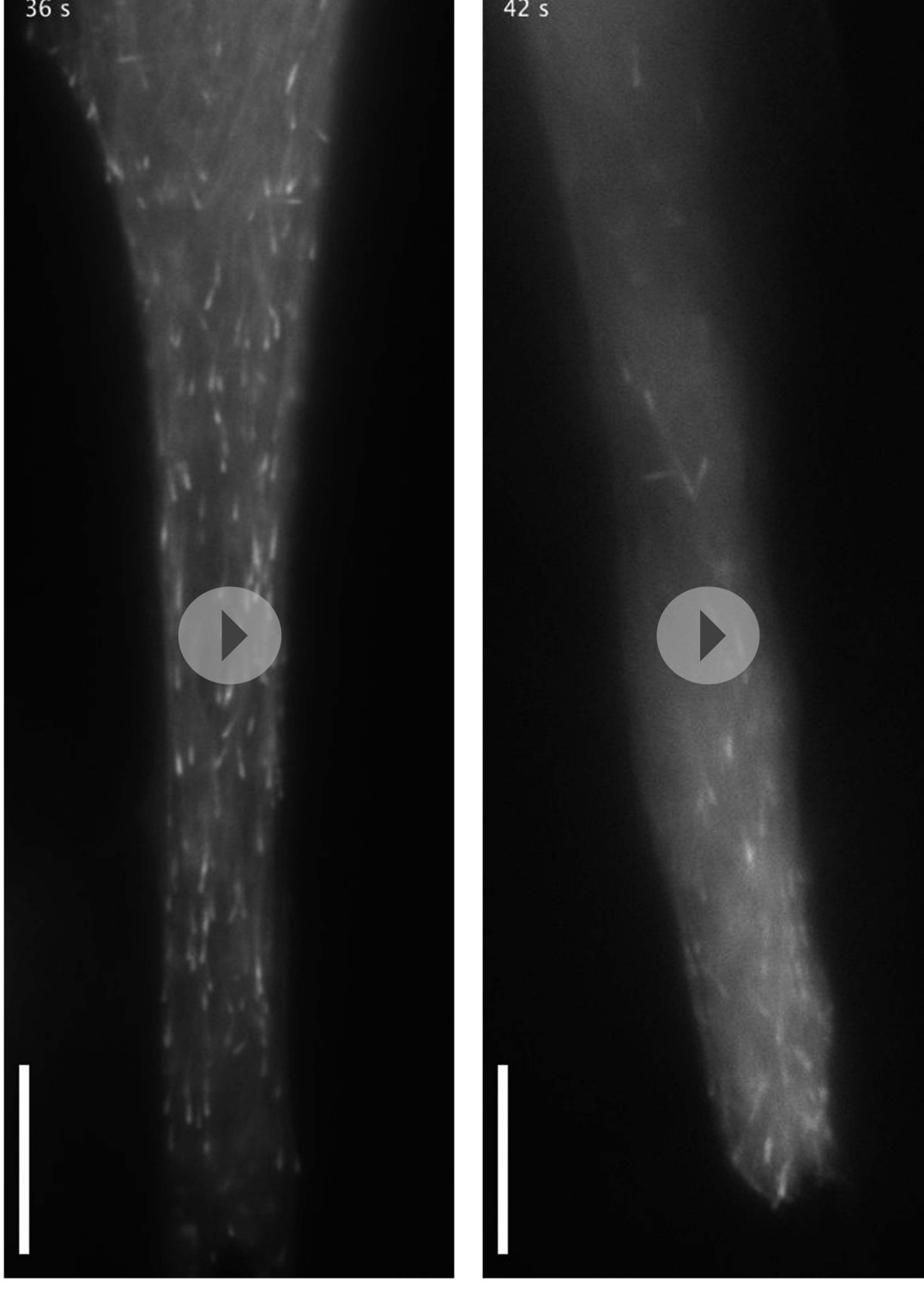

**Video 8.** Growing microtubules in a 48-hr differentiated C2C12 myoblast treated with sh-Control co-expressing EB3-tdTomato. Scale bar: 10 μm.

**Video 9.** Growing microtubules in a 48-hr differentiated C2C12 myoblast treated with sh-oMAP4 co-expressing EB3-tdTomato. Scale bar: 10 μm.

rather than PRC1 itself causing the formation of the overlaps. We do not observe a substantial enrichment of GFP-oMAP4 in antiparallel or parallel overlaps (*Figure 6C*). To determine whether the bias in zippering towards antiparallel-oriented microtubules could be due to differences in the length of

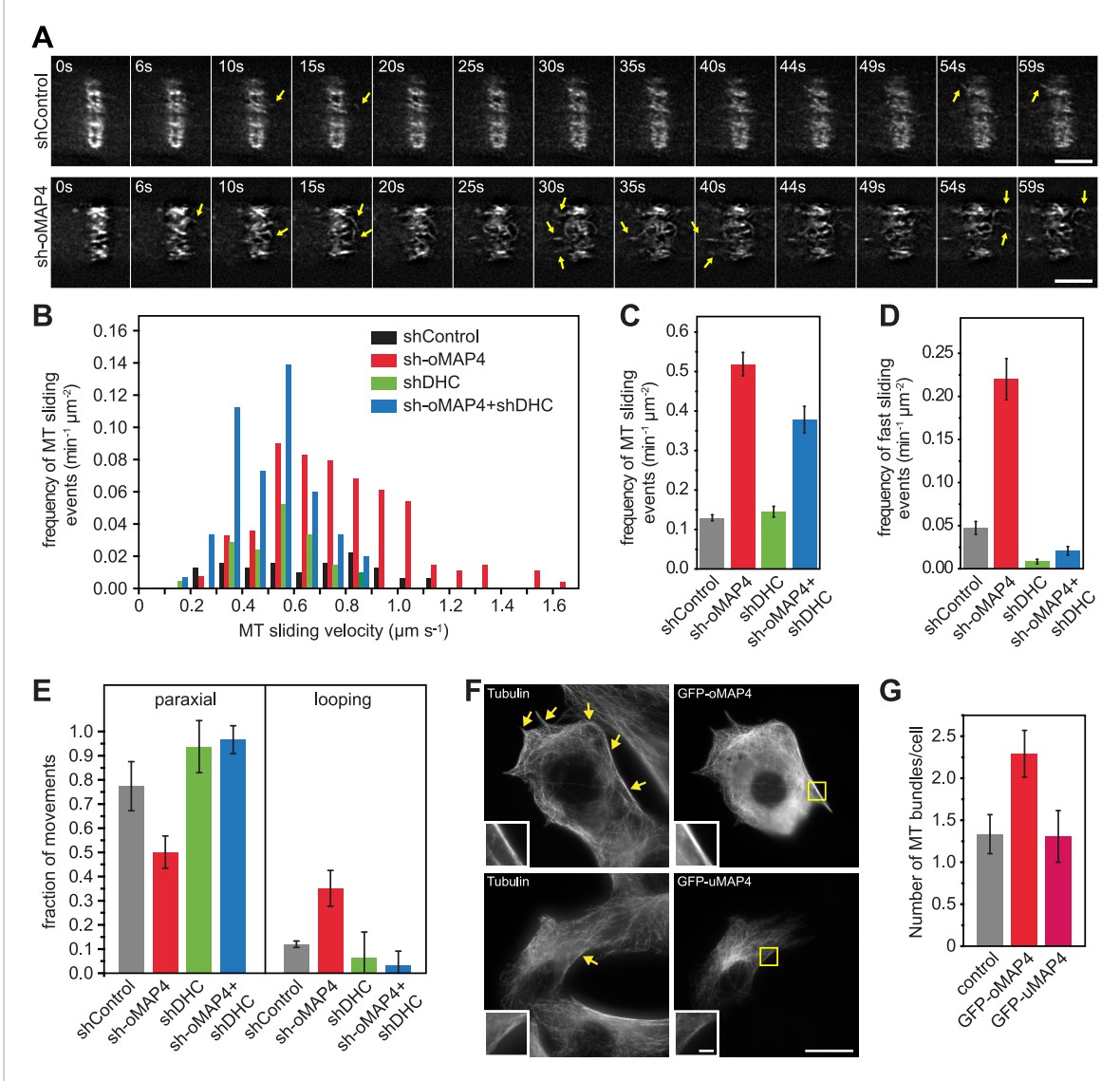

**Figure 4**. oMAP4 prevents microtubule sliding in cells. (**A**) Microtubule motility (arrows) observed after photoconversion of mEOS2-Tubulin in 48 hr differentiated cells treated with shRNA as indicated. Scale bars 5 µm. See supplementary *Videos 10, 11*. (**B**) Frequency of microtubule motility events relative to average sliding velocity in 48-hr differentiated cells treated with shRNA as indicated. Data pooled from three experiments, n = 15–23 cells. (**C** and **D**) Frequency of microtubule motility events: all events are shown in **C**, while only fast sliding events are shown in **D**. The latter were defined as movement faster than 700 nm/s. Data show mean ± SD, n = 3 experiments, 15–23 cells. (**E**) Directionality of microtubule motility events. Paraxial movement is defined as occurring parallel to the long cell axis with a deviation less than 45˚. Looping microtubules changed direction by more than 90˚ during the movement event. Data show mean ± SD, n = 3 experiments, 15–23 cells. (**F** and **G**) Expression of GFP-oMAP4 (upper row) or GFP-uMAP4 (lower row) in undifferentiated C2C12 cells stained with tubulin antibodies. The number of strong microtubule bundles (arrows) of at least 2-µm length and threefold intensity of an individual microtubule was scored. Data in g show mean ± SEM, n = 13–23 cells. oMAP4 causes statistically significant change (p = 0.02). Scale bar 20 µm, 2 µm in insets.

The following figure supplement is available for figure 4:

**Figure supplement 1**. Microtubule orientation in depleted cells.

microtubules that encountered each other, we measured the length dependence of MAP4-mediated zippering. We found similar distributions of microtubule lengths for encounters that resulted in zippering and those encounters that occurred at similarly shallow-incipient angles but not led to zippering (p = 0.22). Likewise, there was no difference between parallel and antiparallel encounters that

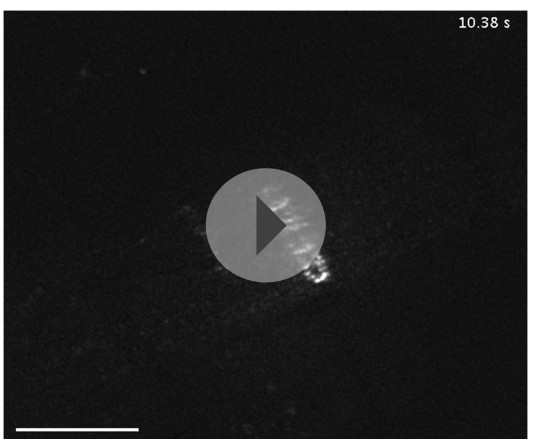
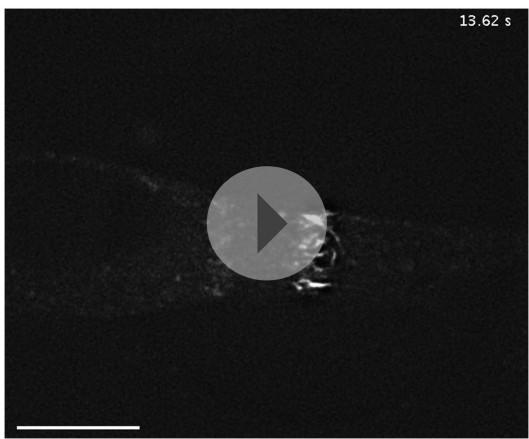

**Video 10.** Photoconversion of bar-shaped patterns of mEOS2-Tubulin in a 48-hr differentiated C2C12 myoblast treated with shControl. Scale bar: 10 μm.

**Video 11.** Photoconversion of bar-shaped patterns of mEOS2-Tubulin in a 48-hr differentiated C2C12 myoblast treated with sh-oMAP4. Scale bar: 10 μm.

were zippered (p = 0.59) and those that did not result in zippering (p = 0.88) (*Figure 6—figure supplement 1*). Therefore, our results demonstrate that oMAP4 is a microtubule-organising factor, which can arrange microtubules into antiparallel and with lesser efficiency parallel bundles. This function is consistent with the depletion phenotypes observed in differentiating muscle cells, where oMAP4 helps to arrange paraxial microtubules and thereby supports cell differentiation.

## oMAP4 bundles withstand motor forces

PRC-1 is dimer that forms highly ordered crosslinks that do not substantially limit microtubule–microtubule sliding at moderate concentrations (*Subramanian et al., 2010*). As we proposed that oMAP4 contributes to microtubule alignment by preventing microtubule sliding in differentiating muscle cells, we next asked how oMAP4-crosslinks microtubules and whether these can withstand motor forces. We considered whether the unique projection domain in oMAP4 confers microtubule cross-linking activity by dimerisation. Proteins of the MAP2/tau family are highly elongated, structurally disordered monomers (*Hernandez et al., 1986*; *Devred et al., 2004*). Using the GOR secondary structure prediction method, we find that oMAP4, uMAP4, and tau share a similar structure of over 60% random coil, about 20% helical, and 13% extended strand (*Garnier et al., 1996*). Furthermore, no significant coiled-coil domain was predicted. To confirm this, we performed sedimentation analysis of bacterially expressed oMAP4 and GFP-oMAP4 in comparison with a number of standard proteins (*Figure 6D,E*). We obtained sedimentation constants of 3.6 ± 0.25 S and 4.3 ± 0.21 S, respectively. Given the molecular weight of the monomers being 99 kD and 131 kD, respectively, we can calculate the frictional ratio $f/f_{min}$ = 2.1, suggesting that oMAP4 is a highly elongated monomer with a shape comparable to tau ($f/f_{min}$ = 1.8, [*Devred et al., 2004*]). We, therefore, predict that oMAP4 most likely bundles microtubules using a second microtubule-binding region in its projection domain.

Single kinesin-1 and dynein molecules can generate forces of up to 7 pN (*Nishiyama et al., 2002*; *Toba et al., 2006*). To determine whether oMAP4 bundling can indeed withstand the forces exerted by several motors tugging the microtubules apart, we performed gliding assays with kinesin-1. The presence of 80 nM oMAP4 did only slightly affect the velocity of kinesin-mediated movements of single microtubules (*Figure 7A,B*). However, antiparallel-microtubule bundles formed in solution and landing on the kinesin surface were often static or their movement was slow and non-persistent (*Figure 7C–E*, *Video 14*). Given their predominantly antiparallel arrangement, forces on the microtubules within a bundle would cancel each other out as long as the linkage between the microtubules is maintained. Occasionally, bundles were driven apart. This usually occurred in bundles where extensive lateral forces were generated on one microtubule that was significantly longer than other microtubules in the bundle (*Figure 7C*). The antiparallel nature of the oMAP4-generated

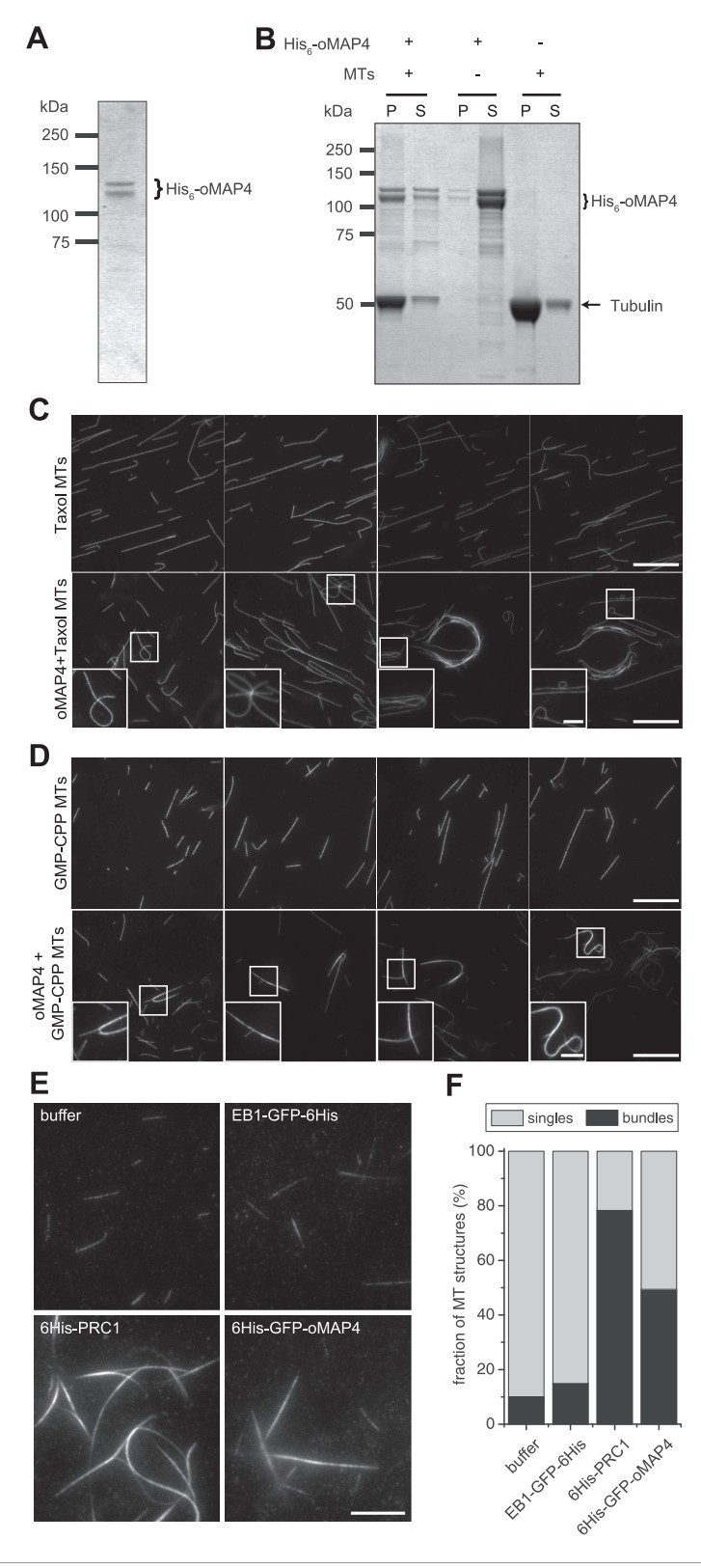

**Figure 5**. oMAP4 bundles microtubules in vitro. (**A**) SDS-PAGE analysis of oMAP4 protein purification. N-terminally 6xHis tagged full-length oMAP4 was purified by Ni²⁺-NTA affinity chromatography followed by ion exchange chromatography. (**B**) Microtubule co-sedimentation analysis of oMAP4 protein. Purified protein from (**A**) was
*Figure 5. continued on next page*

*Figure 5. Continued*

incubated with Taxol-stabilised microtubules prior to centrifugation. Separate centrifugations of microtubules alone and oMAP4 protein alone were used as controls. Pellet and supernatant fractions were analysed by SDS-PAGE and Coomassie staining. (**C** and **D**) Analysis of microtubule bundling by oMAP4 in vitro. Taxol- or GMP-CPP-stabilised microtubules were incubated with buffer or 60 nM purified oMAP4 protein before spreading on glass for imaging. Insets show various microtubule crosslinking and bundling events. Scale bars 20 μm, 5 μm in insets. (**E** and **F**) Comparison of bundling efficiency of oMAP4 with negative control (EB1) and positive control (PRC1). Representative image and data quantified from >300 microtubules are shown. Scale bar 10 μm.

bundles was confirmed in these cases from the opposing direction of their movement after separation (*Figure 7C*, magenta arrows). Most importantly, we observed that more than 75% of microtubule bundles withstand motor forces for the entire duration of our 7.5 min videos (*Figure 7C,E,F*, yellow arrows), suggesting that oMAP4-mediated cross-linking is indeed able to prevent motor-driven microtubule sliding as we hypothesised.

## Discussion

Why is a highly organised microtubule cytoskeleton required to undergo muscle differentiation? Microtubules are the stiffest of the cytoskeletal polymers that can bear high-compressive loads, especially when reinforced laterally (*Brangwynne et al., 2006*). Our data support a model in which microtubules fulfil a structural role during the elongation of muscle cells. If so, one would expect that the degree of alignment of microtubules will correlate with a cell's ability to elongate. We tested this by plotting the Kuiper statistic as a measure of microtubule orderliness against the mean cell length of 48 hr differentiated cells for different RNAi treatments used in this study and found indeed a linear correlation (*Figure 8A*). Likewise, the depletion of mMAP4, which increases the length of differentiating myoblasts also increases the paraxial microtubule alignment beyond that of control cells (*Figure 3—figure supplement 1*). These data support the idea that MT orientation is strongly linked to the morphological changes required for muscle differentiation. As observed for oMAP4, our previously published data on EB3 depletion (*Straube and Merdes, 2007*) and a number of unpublished observations, myoblasts that fail to elongate are also impaired in cell–cell fusion. While we don't yet understand the relationship between cell elongation and fusion, a mechanism to prevent the fusion of myoblasts that did not complete the previous step of differentiation, makes intuitively sense. oMAP4-depleted cells show in addition to impaired morphological changes, also a delay in the expression of myogenic markers, such as myogenin and embryonic myosin, further suggesting that a signalling step in the differentiation programme has not been completed. Microtubules have been implicated in a signalling role during myogenesis based on the observation that altering the level of posttranslational tubulin modifications either through chemical inhibition or depletion of microtubule-stabilising MAPs leads to similar delays in the expression of differentiation markers and impaired formation of myotubes (*Spencer et al., 2000*; *Chang et al., 2002*; *Zhang et al., 2009*). The depletion of oMAP4 does not negatively affect tubulin acetylation (*Figure 3F*) but might impact other aspects of this putative microtubule-dependent signalling event. It will be a future challenge to elucidate the pathway that couples microtubule organisation and chemical modification to the timing of the myogenic protein expression programme.

We show that oMAP4 and PRC1 are both microtubule-bundling proteins, but with very different properties. While PRC1 is an antiparallel dimer that forms ordered crosslinks of a defined distance and specifically enriches in antiparallel-microtubule overlaps (*Bieling et al., 2010*; *Subramanian et al., 2010*), oMAP4 has little preference for binding to bundles, but instead is able to zipper microtubules. Zippering of microtubules growing from surface-attached seeds requires bending of microtubules. We consider that the lateral forces required to bring two microtubules close together against the rigidity of the microtubule have to be borne by a single crosslinking molecule. Thus, our data suggest that single molecules of oMAP4 can resist higher loads than PRC1 crosslinks. When we apply longitudinal forces as experienced by pre-formed bundles in kinesin-gliding assays, the shear forces are probably shared between several crosslinking molecules, thus enabling oMAP4 to withstand counteracting motor forces on the bundled microtubules. Bundles that contain one very long

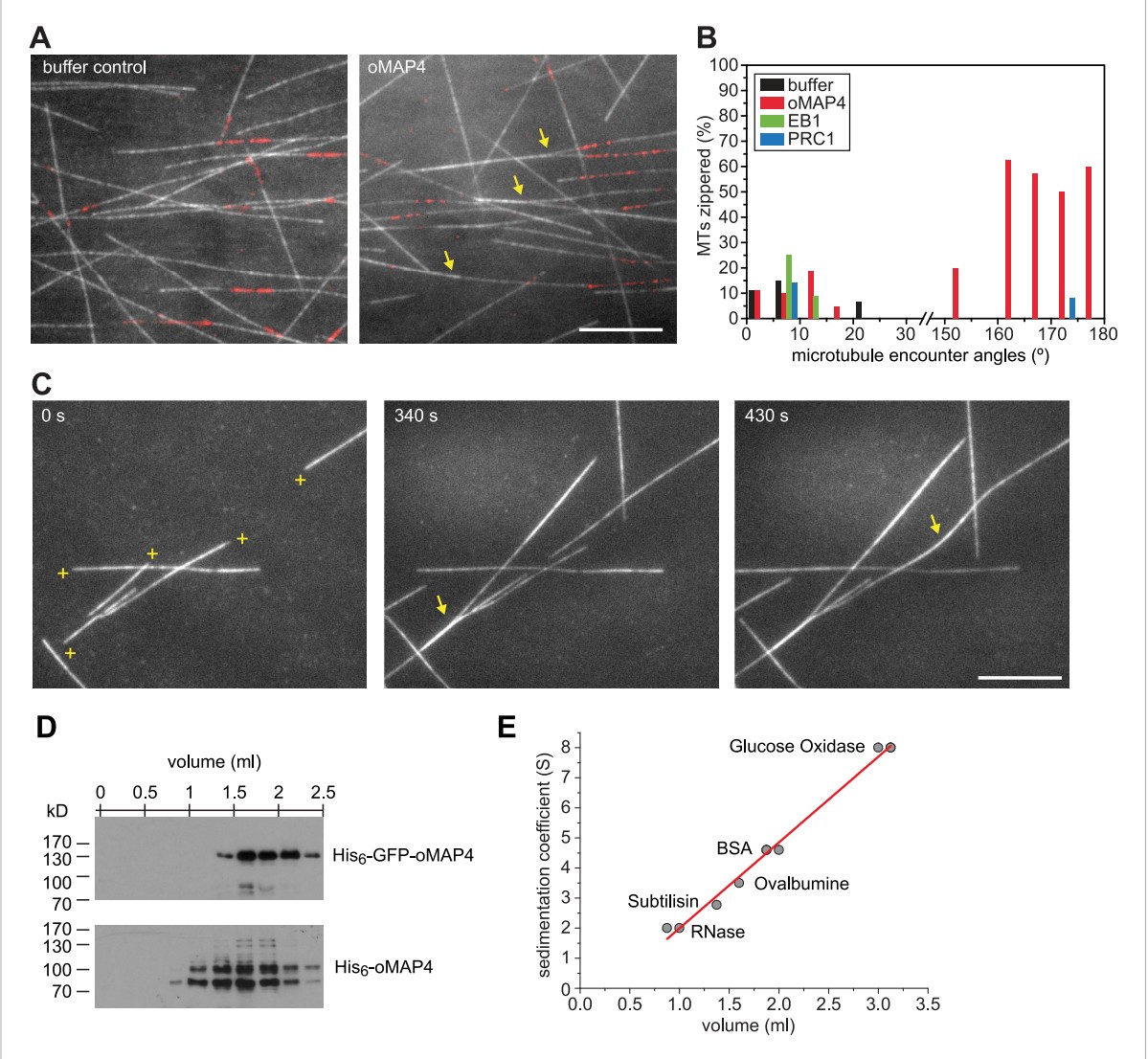

**Figure 6**. oMAP4 zippers dynamic microtubules with a bias for antiparallel arrangements. (**A**) Dynamic Rhodamine-labelled microtubules (greyscale) assembled from immobilised Hilyte640-labelled seeds (red) in vitro are zippered in the presence of oMAP4. Arrows highlight bundled microtubules. Scale bar 10 μm. (**B**) Histogram showing proportion of microtubule encounters leading to MT zippering relative to microtubule encounter angles. Antiparallel encounters are observed at angles above 90°. Note no zippering occurs between 25 and 150°. Data for GFP-oMAP4, EB1-GFP and PRC1, each at 80 nM concentration are shown for comparison. n ≥ 400 microtubule encounters for control and oMAP4, 136 encounters for PRC1 and 184 encounters for EB1. (**C**) Example of zippering events (arrows) in the presence of 80 nM oMAP4-GFP. Microtubule polarity is indicated with (+) at the dynamic plus end. See supplementary *Videos 12, 13*. Scale bar 10 μm. (**D** and **E**) Immunoblot of oMAP4 in fractions from glycerol gradients to determine its sedimentation coefficient with volume from top of gradient indicated (**D**). Calibration curve using standard proteins with known sedimentation coefficients and linear regression curve (**E**).

The following figure supplement is available for figure 6:

**Figure supplement 1**. Microtubule length distribution in zippering experiments.

microtubule that is driven laterally on the kinesin surface, eventually splay apart as lateral forces cannot be shared between the crosslinkers. In contrast, PRC1 cannot resist motor forces and at moderate levels only slightly slows motor-driven sliding of microtubules (*Bieling et al., 2010*; *Subramanian et al., 2010*). Thus, oMAP4 has the required properties to fulfil the role of a microtubule organiser, justifying its name as organising MAP4, although the structural basis for oMAP4's antiparallel zippering remains to be elucidated.

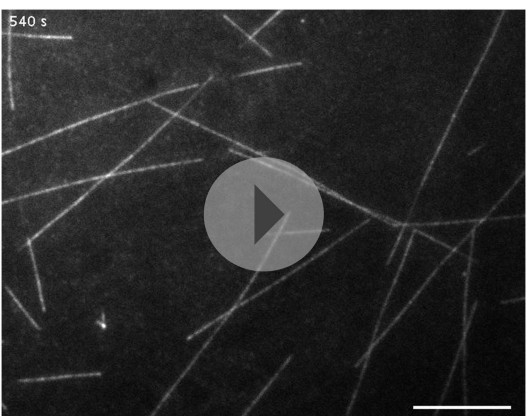

**Video 12.** TIRF-based assay showing dynamic Rhodamine-labelled microtubules assembled from immobilised seeds. Scale bar: 10 µm.

**Video 13.** TIRF-based assay showing dynamic microtubules in the presence of 80 nM GFP-oMAP4. Note that antiparallel microtubule encounters result in zippering into an antiparallel bundle in most cases. Scale bar: 10 µm.

The motors dynein and kinesin move microtubules and cause apparent disorder that is limited by oMAP4 crosslinking. However, both motors have also been reported to bundle and organise microtubules. Indeed, dynein depletion alone results in reduced microtubule alignment (*Figure 4—figure supplement 1*). This is consistent with the finding that dynein controls muscle length in *Drosophila* (*Folker et al., 2012*) and earlier reports of dynein involvement in the self-organisation of microtubule networks and its ability to crosslink and slide antiparallel microtubules as well as transporting microtubules along the cell cortex (*Heald et al., 1996*; *Adames and Cooper, 2000*; *Merdes et al., 2000*; *Fink and Steinberg, 2006*; *Samora et al., 2011*; *Tanenbaum et al., 2013*). As oMAP4 is only able to efficiently zipper microtubules at incident angles of less than 30° if antiparallel and less than 10° if parallel, we propose that dynein-mediated looping and buckling of microtubules (*Figure 4E*; *Fink and Steinberg, 2006*; *Tanenbaum et al., 2013*) brings microtubules into a favourable position for oMAP4-mediated zippering. As oMAP4-mediated bundling resists motor-driven sliding, dynein can only move those microtubules that are not yet aligned to the paraxial network. Thus, dynein and oMAP4 are likely to cooperate in the formation of the highly ordered microtubule arrangement in differentiating muscle cells (*Figure 8C*). In the absence of oMAP4, excessive motor-driven microtubule motility disorganises microtubules. In the absence of dynein, oMAP4 might stabilise high-angle microtubule crossovers, but will not be able to align them into the network. If the activity of both oMAP4 and dynein is reduced, oMAP4 zippering is missing and kinesin-mediated sliding and bundling (*Straube et al., 2006*; *Jolly et al., 2010*) prevails (*Figure 8B*). In agreement with this model, some disorganisation of microtubules has been observed in kinesin-1-depleted myotubes (*Wilson and Holzbaur, 2012*). Microtubule–microtubule sliding has recently been implicated in driving neurite outgrowth (*Lu et al., 2013*), and paraxial microtubule arrangements have been shown to drive dorsal closure during embryonic development (*Jankovics and Brunner, 2006*). Thus, the mechanisms we reveal here for motor and MAP cooperation in the formation of paraxial microtubule networks are likely to be of importance beyond muscle cells. Indeed, oMAP4 is highly expressed in brain (*Figure 2—figure supplement 1*), suggesting that it might be required to support the paraxial microtubule arrays in dendrites and the axon.

While the zippering model described above explains how a paraxial array is maintained in the presence of microtubule turnover, it does not explain how the symmetry of the radial microtubule cytoskeleton in the myoblast is broken in the first place. We think that a bipolar elongating myoblast can be compared to a migrating cell with two fronts. Symmetry breaking in cell migration occurs through protrusion mediated by the actin cytoskeleton. Microtubules support and stabilise cell protrusions, and microtubule plus ends are selectively stabilised at the leading edge (*Waterman-Storer et al., 1999*; *Kaverina and Straube, 2011*). This establishes and reinforces the new polarity axis. Similarly, in differentiating myoblasts dynamic capture of microtubule plus ends occurs at the

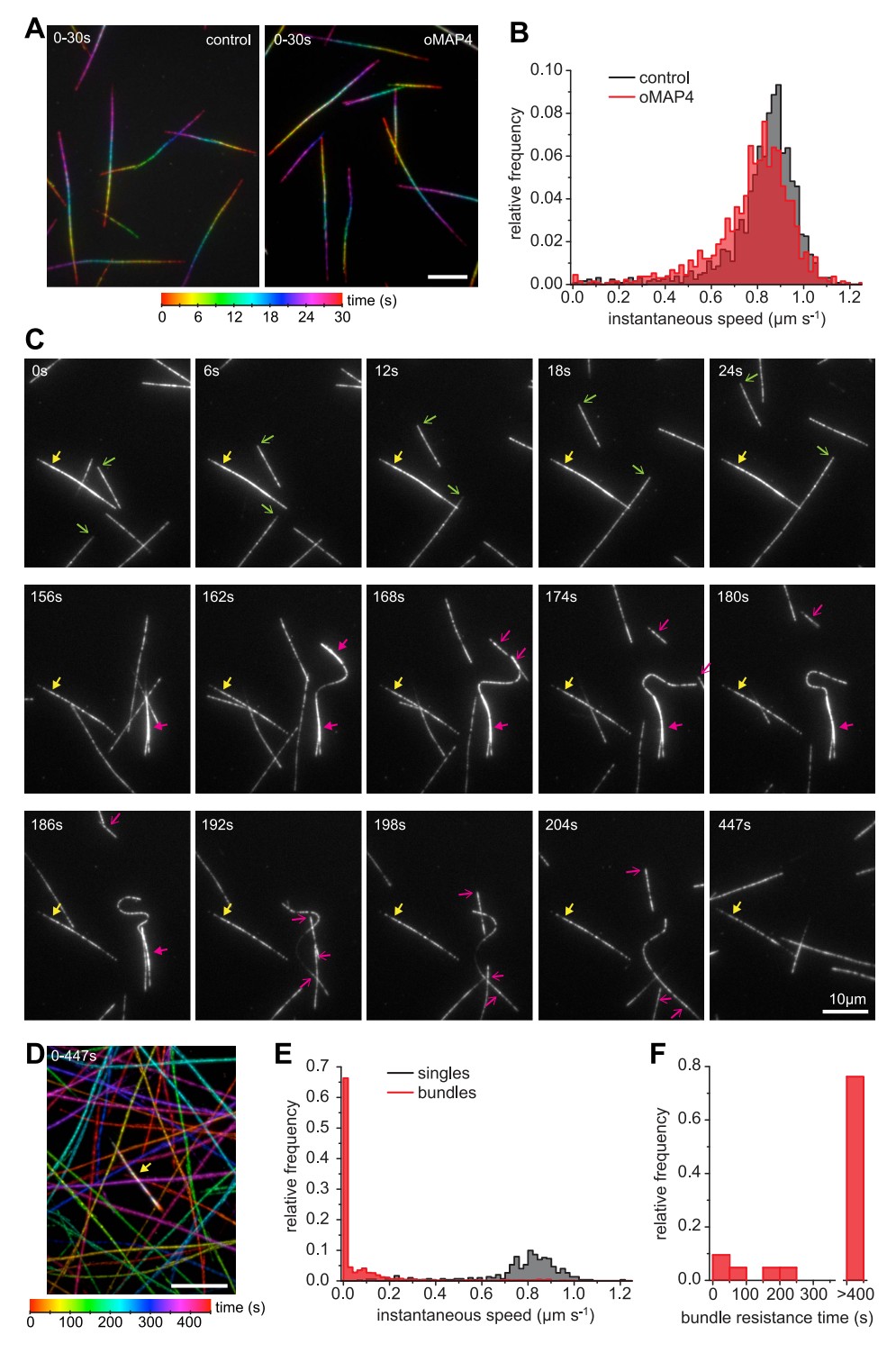

**Figure 7**. oMAP4 bundles can withstand motor forces. (**A**) Microtubule gliding assay on a Drosophila kinesin-1-coated surface. Time colour-coded projections over 30 s are shown for buffer control and 80 nM oMAP4. (**B**) Instantaneous speeds of microtubule motility were determined from tracks of 50 microtubules and shown as histograms. The presence of 80 nM oMAP4 reduces velocity of single microtubules by about 5%. (**C**) Microtubule bundles formed in solution by oMAP4 and landing on the kinesin-coated surface tend not to move (yellow filled arrows), while single microtubules do (green arrows). In about 25% of cases, bundles are driven apart by motor forces (magenta filled arrows) with microtubules gliding apart at normal speed once separated (magenta arrows).
*Figure 7. continued on next page*

*Figure 7. Continued*

See supplementary *Video 14*. (**D**) Time colour-coded projection of gliding assay showing a microtubule bundle (arrow, appears white due to averaging of all time points) and individual microtubules over 447 s in the presence of 80 nM oMAP4. (**E**) Instantaneous speeds of microtubule motility were determined from tracks of 20 single and bundled microtubules and shown as histograms. (**F**) Time between start of observation and either end of observation or time when bundle was separated by motor forces. Bundles that moved out of the field or dissociated as intact bundle were not scored.

cell tips (*Straube and Merdes, 2007*). This leads to the selective stabilisation of paraxial microtubules to break the symmetry in the microtubule network. The zippering mechanism described here would promote shorter microtubules to align to longer and already bundled microtubules and thus amplify the directional bias and increase stability and orderliness in the paraxial array. It is likely that the guidance of microtubule growth through the cooperation of plus end tracking proteins and motors also contributes to maintenance of the ordered array (*Mattie et al., 2010*). It will be interesting to dissect the contributions of these different pathways to microtubule network reorganisation, not only at the onset of myogenesis, but also during later stages when the parallel network is replaced by the grid-like lattice found in adult muscle (*Oddoux et al., 2013*).

## Materials and methods

### RNA extraction, cDNA preparation, and gene expression analyses

RNA was extracted from 0 to 60 hr differentiated C2C12 cells using Trizol reagent (Invitrogen, Life Technologies, UK), and random-primed cDNA was synthesized using RevertAid H Minus M-MuLV Reverse Transcriptase (Fermentas, Fisher Scientific, UK) according to manufacturer's protocol. PCR reactions were carried out using cDNA time-course samples as templates, isoform-specific upstream primers GCCAGCCTTCTGAGCCTTG (for uMAP4), GAGATCCAAGATGTTCAAGTC (for mMAP4), and CTGTTGGAAGAGACCCCAC (for oMAP4) and downstream primers CAGCTGGCACTGAGCCTG (to determine relative expression levels) and GAAGGGCCTCACTGCCAC (to determine number of microtubule binding repeats). As control, the coding sequence of glyceraldehyde-3-phosphate dehydrogenase (GAPDH) was amplified using the primers CCCACTTGAAGGGTGGAG and CAGGCGGCACGTCAGATC.

For differential expression analyses of MAP4 isoforms, the mouse genome assembly (release date July 2007 (NCBI37/mm9)) was examined using the UCSC Genome Browser at http://genome.ucsc.edu/. For RNA sequencing analyses, raw data from previously described C2C12 data sets (*Trapnell et al., 2010*, see *Supplementary file 1*) of the Mouse ENCODE project were exported. Reads per million (RPM) values were then summed and divided by the length of sequenced regions to obtain RPM/Kb values. Expression levels of MAP4 isoforms in various tissues were compared by exporting and analysing published Affymetrix mouse exon array data sets (*Pohl et al., 2009*, see *Supplementary file 2*) from the Genome Browser.

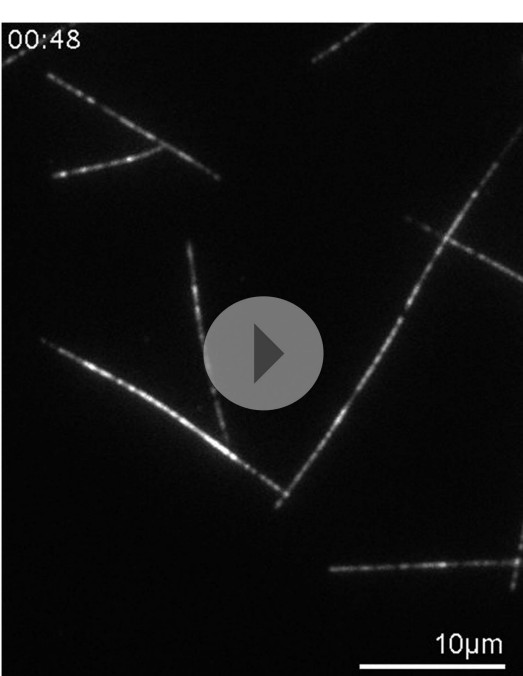

**Video 14.** Microtubule gliding assay on a kinesin-1-coated surface in the presence of 80 nM GFP-oMAP4. Note that single microtubules move persistently, while bundles don't until they are driven apart. Scale bar: 10 μm.

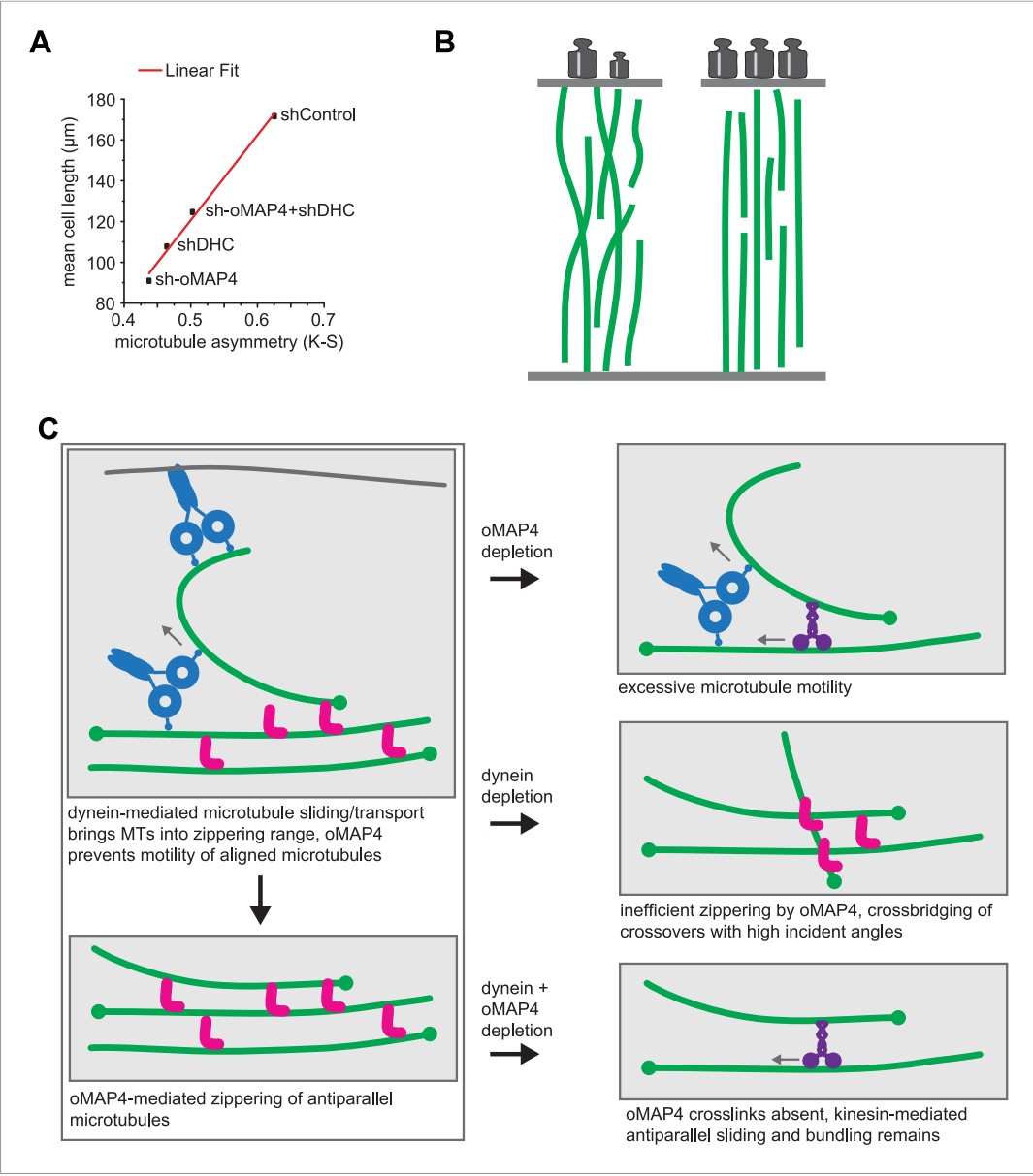

**Figure 8**. oMAP4 and dynein co-operate in the organisation of the paraxial microtubule network in differentiating muscle cells. (**A**) Kuiper statistics of traced microtubule filaments as in *Figures 3G, 4*, *Figure 4—figure supplement 1* as a measure for microtubule orderliness is plotted against the mean cell length of 48 hr differentiated C2C12 myoblasts treated with shRNAs as indicated. A linear fit to the data is shown. (**B**) We propose that the correlation between the precision of paraxial alignment and cell elongation suggests that ordered microtubule arrays confer higher mechanical stability to counteract contractile forces. (**C**) Model of cooperation of oMAP4-mediated zippering with motor-driven microtubule sliding/transport in the formation of a highly ordered paraxial microtubule network. Microtubules are indicated in green, oMAP4 in magenta, dynein in blue, and kinesin in purple.

## Cloning of MAP4 isoforms, shRNA plasmids, mammalian and bacterial expression constructs

For cloning uMAP4 (accession number M72414) and oMAP4 (accession number BC042645), cDNA was amplified using primers CAGGTCGACAGAATGGCCGACCTCAG and GACCGCGGACGAGAC CAGAATGTCATC for uMAP4, and GAGGTCGACATGGACTCCCGGAAAGAAATC and CCCG CGGTCTCAATTTGTCTCCTGG for oMAP4. PCR products were cloned into the MscI site of pCAP$^s$

(Roche, UK) and confirmed by sequencing, digested with SalI and SacII, and transferred into pEGFP-C1 (Clontech, France). To clone mMAP4 (assembled from sequences under accession numbers M72414 and U08819), two overlapping fragments encoding its N-terminal and C-terminal domains were generated by PCR using primers CAGGTCGACAGAATGGCCGACCTCAG and CTGCAAT CAGCAAGCCCAC, and CCTCTGGGAGATCACCATC and CCCGCGGTCTCAATTTGTCTCCTGG, respectively, and each cloned into pCAP^s and sequenced. Plasmids carrying N-terminal and C-terminal coding sequences were then digested with SalI + SpeI and SpeI + SacII, respectively, and cloned into the SalI + SacII sites of pEGFP-C1. To clone FLAG-oMAP4, a 5X-FLAG tag (DYKDADLDKDDDDK) was amplified by PCR and digested with NgoMIV and SalI. It was then inserted into pEGFP-oMAP4 that was opened with AgeI and SalI, thereby replacing the eGFP ORF. FLAG-oMAP4^RIP was generated using FLAG-oMAP4 as a template and primers GAGGTCGACATG GACTCCCGGAAAGAAATC, CTCCTCGAGTTAAGCAGTTGGTACCTGAG, and CCCAGCTGTGAAC TTGGTTGATAAGTACCCCTG in 3-step mutagenesis PCR.

Bacterial expression vector for N-terminally 6xHis-tagged oMAP4 was generated by SalI + MfeI digestion of pEGFP-oMAP4 (four MTB repeats) and insertion of the coding sequence into the XhoI + EcoRI sites of pRSET-A. N-terminally 6x-His-eGFP-tagged oMAP4 expression construct was cloned by transferring EGFP-oMAP4 with NcoI-MfeI from pEGFP-oMAP4 (five MTB repeats) into the NcoI-EcoRI sites of pRSET-B vector (Invitrogen).

To clone GST fusions of MAP4 fragments for polyclonal antibody generation, regions specific to mMAP4 (amino acids 631–1060) and oMAP4 (amino acids 22–350) were obtained by PCR using pEGFP-mMAP4 (primers GCCAGCCTTCTGAGCCTTG and CTCCTCGAGTTAAGTAATGGCCCCT GGTTG) and pEGFP-oMAP4 (primers GAGGGTCAATTGAATGAAATCGGGCTGAATG and CTCCTC GAGTTAAGCAGTTGGTACCTGAG). mMAP4 and oMAP4 PCR products were then digested with EcoRI + XhoI and MfeI + XhoI, respectively, and inserted into the EcoRI + XhoI sites pGEX-6P-1 vector (GE Healthcare, UK).

Depletion constructs were based on pSUPERneo.gfp (Oligoengine, Seattle, WA). shRNA-target sequences were chosen using Oligo Retriever (http://cancan.cshl.edu/RNAi_central/RNAi.cgi?type= shRNA) and were as follows: shControl (targeting Luciferase): CGTACGCGGAATACTTCGA; sh-uMAP4: GCCTTGCTCAGGAGTATCC; sh-mMAP4: CAGAGAGTTTGGATAAGAA; sh-oMAP4: GCTGTG AATCTTGTCGATAAG; shDHC: CAATTACAGTCTGGAGTTA; shKHC (targeting Kif5b): CAATTGGAGT TATAGGAAA. The neo.gfp cassette was replaced by GFP-Tubulin, mCherry-Tubulin, or mEos2-Tubulin.

EB3-tdTomato, paGFP-Tubulin, and mEos2-Tubulin were described previously (*Rusan and Wadsworth, 2005*; *McKinney et al., 2009*; *Samora et al., 2011*).

## Generation and purification of polyclonal antibodies

Rabbit polyclonal antibodies against specific regions in oMAP4 and mMAP4 were raised against affinity purified GST-oMAP4$_{22-350}$ and GST-mMAP4$_{631-1060}$, respectively by Absea Biotechnology Ltd. (China). Polyclonal antibodies were purified from sera using the same GST fusion proteins. To do this, purified protein samples were run on SDS-PAGE, transferred to nitrocellulose membranes, protein bands were stained with Ponceau S and excised using a sterile scalpel. Membrane pieces were blocked with 5% wt/vol milk in TBST, cut into small pieces and incubated over night at 4°C in 1 ml TBST and 1 ml antiserum. Membrane pieces were washed thrice with TBST and antibodies were eluted by adding 200 µl 100 mM Glycine-HCl (pH 2.5) and vortexing for 1 min. Purified antibodies were then transferred to fresh tubes and neutralised immediately by adding 10 µl 1 M Tris-Base. Antibodies were stored at −20°C after adding Glycerol to a final concentration of 50% vol/vol.

## Cell culture, plasmid transfections, and myoblast differentiation

Mouse C2C12 myoblasts were cultured on rat tail collagen (Sigma, UK) in DMEM-GlutaMAX (Invitrogen) supplemented with 10% fetal bovine serum (FBS), 2 mM L-Glutamine, 100 U/ml penicillin, and 100 µg/ml streptomycin with 5% $CO_2$ in a humidified incubator. For localisation analyses, myoblasts were transfected with 1 µg plasmid DNA using Fugene 6 reagent (Roche) and analysed 48 hr post-transfection. For myoblast elongation and fusion analysis, 10,000 or 20,000 C2C12 cells, respectively, were seeded onto collagen-coated coverslips. Cells were transfected 24 hr later with a total amount of 1.5 µg shRNA constructs or shRNA and rescue constructs using Lipofectamine Plus reagent (Invitrogen) in OptiMEM (Invitrogen), which was replaced with growth medium 4–6 hr after transfection. Muscle cell differentiation was induced by replacing growth medium with differentiation

medium (DMEM, 0.1% FBS, 2 mM L-glutamine, 5 µg/ml insulin, 5 µg/ml, transferrin, 100 U/ml penicillin and 100 µg/ml streptomycin) 20 hr after shRNA transfection. DMEM was changed daily and analysis of shRNA transfected cells was consistently performed 72–78 hr after transfection.

## Live cell imaging and immunofluorescence experiments

Live cells were imaged at 37°C and 5% $CO_2$ in a stage top incubator (Tokai Hit, Fujinomiya, Japan) using a 100× or a 60× oil NA 1.4 objective on a Deltavision system (Applied Precision, LLC, Issaquah, WA) using Chroma filter sets and a Coolsnap HQ camera controlled by SoftWorx (Applied Precision, LLC). For all analysis of microtubule phenotypes, mono-nucleated, elongated cells were selected. For analyses of microtubule dynamics during myoblast differentiation, fluorescence images of EB3-tdTomato were acquired with 500 ms exposure at a temporal resolution of 3 s for 120 s. EB comets were tracked using plusTipTracker and further analysed for microtubule directionality using custom MATLAB code (see supplementary file MTdirectionality.m). For microtubule motility experiments, cells were transiently transfected with pSuper-mEOS2-Tubulin plasmids or co-transfected with paGFP-Tubulin and pSuper-mCherry-Tubulin plasmids. The microtubule cytoskeleton was focussed and bar-shaped regions of interest perpendicular to the microtubule network were selected using the non-converted mEOS2-Tubulin or mCherry-Tubulin signal. After photoactivation or photoconversion of these regions of interest using 10% 406 nm with the Deltavision photokinetics module laser power, images were acquired with 500 ms exposure at a temporal resolution of 1.6 s per frame for 60 s or lower temporal resolution for up to 6 min. Images were deconvolved with low noise filtering method for 10 iterations using SoftWorx (Applied Precision, LLC). Microtubule motility events were scored when an activated fragment moved for more than 0.5 µm from its original location. The velocity of microtubule movements was determined as the average velocity during each motility event, which lasted on average 15–20 s. Directionality of microtubule motility was scored as paraxial if within 45° of the long axis of the cell. Looping microtubules were defined as those that underwent a directional change of more than 90° during the movement event. For analysis of the contribution of microtubule dynamics and motor-driven microtubule movement to microtubule loss from photoactivated regions, undifferentiated or 48 hr differentiated C2C12 cells expressing mEOS2-Tubulin were treated with either 10 µM Taxol (to suppress MT depolymerisation) or 5 mM azide in 1 mM 2-deoxyglucose (to suppress motor activity) or a mixture of Taxol and azide (as a control for photobleaching) for 30 min before imaging. Signal intensity of photoconverted regions at each time point recorded using ImageJ. After background subtraction, photobleaching correction and normalisation of fluorescence intensity, the half-life of signal dissipation was determined by fitting a single exponential curve to the fluorescence intensity data over time.

For immunofluorescence experiments, cells were fixed in −20°C cold methanol for 24–48 hr. Fixed cells were rehydrated by washing with PBS for 5 min and blocked with 0.5% BSA wt/vol in PBST for 5 min. Coverslips were then incubated with primary antibodies overnight at 4°C or for 4 hr at RT and secondary antibodies for 1 hr in 0.5% BSA wt/vol in PBST. To stain DNA, coverslips were incubated with 4′,6-Diamidino-2-phenylindole dihydrochloride (#D9542; DAPI; Sigma) for 2 min. Coverslips were then mounted using Vectashield mounting medium (#H-1000; Vector Labs, UK). Primary antibody dilutions were as follows: rabbit anti-mMAP4 (1:1000), rabbit anti-PCM-1 (1:500, *Dammermann and Merdes, 2002*), mouse anti-embryonic myosin (1:50; F1.652, DSHB, University of Iowa), mouse anti-α-tubulin (1:1000; DM1A, Sigma), mouse anti-acetylated tubulin (1:1000; 6-11B-1, Sigma), mouse anti-myogenin (1:50; F5D, Santa Cruz (Dallas, TX)). Alexa488-, Alexa594- or Alexa647-conjugated anti-mouse or anti-rabbit secondary antibodies (Molecular Probes, Life Technologies, UK) were used at 1:500 dilution. For myoblast elongation analysis, cells were imaged using a 20× objective on the Deltavision system as above. Cell lengths of GFP-Tubulin-positive mono-nucleated cells were measured as straight-line distance from tip-to-tip using Image-Pro Analyzer 7.0 (Media Cybernetics, UK). For cell fusion analysis, cells with two or less nuclei (visualised as holes in GFP-Tubulin signal) and those with 3 or more nuclei were counted on the entire coverslip directly on the Deltavision microscope using a 20× objective and GFP filter sets. To analyse microtubule filament arrangement, GFP-Tubulin was imaged on a Perkin–Elmer UltraView spinning disk confocal system using a 488 nm laser and an Orca-R2 camera (Hamamatsu, Japan) under the control of Volocity software (Perkin–Elmer, Waltham, MA). Filaments were manually traced using ImageJ and deviation from the longitudinal cell axis calculated in MATLAB. To determine the number of bundles per cell, undifferentiated C2C12 cells were transiently transfected with either GFP-oMAP4 or GFP-uMAP4,

fixed after 24 hr and stained with tubulin antibodies. Bundles were scored when elongated regions of more than 2.3-µm length had an intensity of more than threefold of a single microtubule. Superresolution images of microtubules labelled with anti-tubulin and Alexa488-conjugated anti-mouse antibodies and embedded in ProLong Gold reagent (Molecular Probes) were obtained on a N-SIM system (Nikon, Japan) at The Babraham Institute using 3D SIM mode with 15 individual images collected for reconstruction.

## Immunoblotting

For validation of depletion, shRNA-transfected and 52–54 hr differentiated cells were detached from culture dishes with Trypsin- EDTA. Trypsinised cells were then transferred to PBS +2% FBS and GFP expressing cells were sorted on FACSDiva or Influx instruments (BD Biosciences, UK). Collected cells were gently pelleted at 300×$g$ for 2 min, resuspended to 10,000 cells/µl in 1× sample buffer and incubated at 95℃ for 5 min. Immunoblotting was performed as described previously (*Samora et al., 2011*). Primary antibody dilutions were as follows: mouse anti-α-tubulin (1:10,000; DM1A, Sigma), mouse anti-FLAG (1:5000; FLAG-M2, Sigma), mouse anti-DHC (1:500; R-325, Santa Cruz), mouse anti-uKHC (1:750; H-50, Santa Cruz), rabbit anti-uMAP4 (1:1,000, H-300, Santa Cruz), rabbit anti-mMAP4 (1:5000), rabbit anti-oMAP4 (1:5000), mouse anti-pentahis (1:3000, Qiagen (UK)). Horseradish peroxidase conjugated anti-mouse or anti-rabbit secondary antibodies (Promega, UK) were used at 1:4000 dilution.

## Protein purification and glycerol gradients

6xHis-oMAP4 and 6xHis-GFP-oMAP4 were expressed in *E. coli* strain BL21-CodonPlus-(DE3) and expression was induced with 0.5 mM isopropyl-β-D-thiogalactoside at 37℃. Bacteria were lysed in binding buffer (50 mM NaPO$_4$ buffer, pH 8.0; 300 mM NaCl; 2 mM β-mercaptoethanol; 15% glycerol) by sonication. oMAP4 was bound to Ni-NTA resin (Qiagen) and eluted with 250 mM imidazole in binding buffer. After twofold dilution with low-salt buffer (20 mM MES, pH 6.8; 1 mM EGTA; 0.5 mM MgCl$_2$) the MAP4-containing fractions were loaded on SP Fast Flow Sepharose (GE Healthcare), washed with low-salt buffer and eluted with a step gradient of high-salt buffer (20 mM MES, pH 6.8; 1 mM EGTA; 0.5 mM MgCl$_2$; 1 M NaCl). All purification was carried out at room temperature to minimise protein aggregation. Proteins were analysed by SDS–polyacrylamide gel electrophoresis and SimplyBlue staining (Invitrogen). Buffer exchange to BRB80 (80 mM PIPES, pH 6.8; 1 mM MgCl$_2$; 1 mM EGTA) was carried out using Vivaspin spin columns (Sartorius, Germany) according to the manufacturer's protocol. GST fusion proteins for antibody generation and purification were bound to Glutathione-Agarose (Sigma), washed with PBS and eluted with GST elution buffer (50 mM Tris-HCl, pH-8.0; 150 mM NaCl; 2 mM 2-Mercaptoethanol; 10 mM Glutathione). Full-length *Drosphila* kinesin-1 was purified previously (*Braun et al., 2009*). EB1-GFP-6xHis was purified as described previously (*Grimaldi et al., 2014*). 6xHis-PRC1 was purified using Ni-NTA as described for 6xHis-oMAP4 above.

To determine sedimentation properties, cleared extracts of BL21 cells expressing 6xHis-oMAP4 and 6xHis-GFP-oMAP4 in BRB80 plus complete protease inhibitors (Roche) were loaded onto 5-ml 10–40% vol/vol glycerol gradients prepared with a Gradient Master (Biocomp, Canada) and spun at 45,000×$g$ in a SW55Ti rotor for 14 hr. Standard proteins were loaded at 5 mg/ml individually on separate gradients. Gradients were fractionated by pipetting from the top in 250 µl aliquots and analysed by measuring the absorbance at 280 nm or analysing band intensity on coomassie-stained polyacrylamide gels or immunoblots using pentahis antibodies (Qiagen). The frictional ratio was determined $f/f_{min} = S_{max}/S$ with $S_{max} = 0.00361 \cdot M^{2/3}$ in Svedbergs for a protein of mass $M$ in Daltons (*Erickson, 2009*).

## In vitro microtubule zippering and kinesin gliding assays

Tubulin was prepared from pig brains according to published protocols (*Gell et al., 2011*). Labelled tubulin was from Cytoskleleton Inc. (Denver, CO), nucleotides were from Jena Biosciences (Germany) and all other chemicals were from Sigma unless indicated. Microtubule seeds were assembled from tubulin, biotin-tubulin, and Hilyte647-tubulin at a molar ratio of 25:1:2 in the presence of 1 mM GMP-CPP in MRB80 (80 mM PIPES, pH 6.8 with KOH, 1 mM EGTA, 4 mM MgCl$_2$) for 1 hr at 37℃, diluted 20-fold with MRB80 + 2 µM Paclitaxel and stored at RT. A 100-µm deep flow chamber was made from a slide and a hydrochloric acid-treated coverslip using double-sided tape (Scotch 3M, UK). For MT dynamics assays, the flow chamber was passivated with PLL-PEG-50% biotin (Susos AG, Zurich, Switzerland). Seeds were attached to this surface using streptavidin and blocked with 1 mg/ml κ-casein. A reaction mix containing 15 µM tubulin, 1 µM X-Rhodamine tubulin, 50 mM KCl, 1 mM GTP,

0.6 mg/ml κ-casein, 0.2% methyl cellulose, 4 mM DTT, 0.2 mg/ml catalase, 0.4 mg/ml glucose oxidase, 50 mM glucose in MRB80, supplemented with 80 nM GFP-oMAP4 or buffer was clarified for 8 min at 190,000×g in an airfuge (Beckman Coulter, UK), the supernatant added to the flow chamber and sealed with candle wax. For gliding assays, 3 nM *Drosophila* kinesin in MRB80, supplemented with 10 mM β-Mercaptoethanol and 0.1 mM MgATP were flown into the glass chamber, before blocking with 1 mg/ml κ-casein. The gliding mix containing X-rhodamine- and Hilyte647-labelled microtubules, 1 mM ATP, 4 mM MgCl$_2$, oxygen scavenger system (4 mM DTT, 0.2 mg/ml catalase, 0.4 mg/ml glucose oxidase, 50 mM glucose), ATP regeneration system (5 mM phosphocreatine, 7 U/ml creatine phosphokinase), 80 nM GFP-oMAP4 in MRB80 was added to the flow chamber and sealed with candle wax. Microtubule assembly and gliding assays were observed on an Olympus TIRF system using a 100× NA 1.49 objective, 1.6× additional magnification, 488 nm, 561 nm, and 640 nm laser lines, a Hamamatsu ImageEM-1k back-illuminated EM-CCD camera under the control of xcellence software (Olympus, Germany). Microtubule gliding speeds were determined from tracks obtained using ImageJ plugin MTrackJ. Microtubule encounters were classified as zippering when they resulted in bundling for a minimum length of 2 µm away from the initial point of contact. For parallel bundling, zippering would need to occur towards the minus end of both microtubules. For antiparallel zippering, both aligned microtubule ends would need to continue growth for at least 2 µm along the lattice of the other microtubule. Microtubule lengths were measured to address whether the bias in microtubule zippering by oMAP4 arises from differences in lengths between parallel and antiparallel microtubules. For microtubule encounters in either orientation that resulted in successful zippering, distance from the point of encounter to the microtubule seed or to the next microtubule crossover point was measured. Microtubule lengths were measured similarly for unsuccessful encounters but these were restricted to shallow angle encounters at which oMAP4-mediated zippering typically occurs, that is, 10˚–30˚.

## Microtubule pelleting and bundling assays

Microtubules were polymerised in BRB80 buffer from 40 to 50 µM pig-brain tubulin in the presence of 1 mM GTP by incubation at 37˚C for 30 min and stabilised by addition of 20 µM Paclitaxel (Sigma). To remove non-polymerised tubulin, microtubules were pelleted by centrifugation at 45,000 RPM for 25 min at 27˚C, washed with and resuspended in BRB80 and 20 µM Paclitaxel. To test binding of MAP4 proteins to microtubules, 1.5 µM of Taxol-stabilised microtubules was mixed with purified 60 nM oMAP4 in the presence of 20 µM Paclitaxel in BRB80 in a total volume of 50 µl. Reactions were then incubated for 20 min at 37˚C and centrifuged at 35˚C, 50,000 RPM for 15 min. After recovery of the supernatant, the pellet was washed with and then resuspended in 50 µl BRB80 and 20 µM Paclitaxel. After addition of SDS-PAGE sample buffer, pellet and supernatant fractions were analysed by SDS-PAGE. For microtubule bundling experiments, 40 µl drops of 0.1 mg/ml Poly-L-Lysine (Sigma) were placed onto 70% ethanol washed and air-dried 22 × 22 mm glass coverslips (Menzel-Gläser, Germany). After removal of the Poly-L-Lysine solution using a bench top coverslip centrifuge (Technical Video Ltd., Port Townsend, WA), coverslips were rinsed three times with 150 µl ddH$_2$O using the same procedure and air dried. To test if oMAP4 can bundle microtubules, Rhodamine-labelled microtubule seeds stabilised with GMP-CPP or Taxol-stabilised microtubules were mixed in BRB80 in a total volume of 10 µl and incubated with buffer or with 60 nM oMAP4 at 22˚C for 10–15 min. 1 µl samples of reactions were then placed onto slides, mounted with Poly-L-Lysine coated coverslips, and imaged using the Deltavision widefield system as above. For comparative analysis of EB1-, PRC1-, and oMAP4-mediated bundling, freshly polymerised GMPCPP-stabilised Hilyte647-labelled microtubules were incubated with 80 nM protein in MRB80 for 10 min before loading into a 10 µm flow chamber made from a slide and a poly-lysine-coated coverslip. At least 30 random fields were imaged using TIRF microscopy at critical angle. Bundles were counted if at least three times as bright as single microtubules.

## Statistical data analysis

Statistical data analyses and graphing were performed using Origin Pro 8.5 (OriginLab, Northampton, MA) or MATLAB (Mathworks, Natick, MA). Image preparation for publication was done using deconvolution in Softworx (Applied Precision), ImageJ and Adobe Illustrator. All statistical significance analyses were carried out using two-tailed two-sample t-tests assuming equal variance. Kuiper statistics was calculated from cumulative distributions of data compared to a random distribution using custom MATLAB code (see *Source code 1* MTdirectionality.m).

## Acknowledgements

We thank Pat Wadsworth (UMass, Amherst) for paGFP-Tubulin, Loren Looger (Janelia Farm) for mEos2-Tubulin, Stephen Royle (University of Warwick) for pRSET-PRC1, Andrew McAinsh (University of Warwick) for purified *Drosophila* kinesin, Ben Fitton for purified EB1-GFP, Ian Titley (ICR, London) and Muriel Erent (University of Warwick) for help with cell sorting. We are grateful to Rob Cross and Masanori Mishima for critical comments on the manuscript. This work was supported by funding from Marie Curie Cancer Care, the University of Warwick and the Lister Institute of Preventive Medicine to AS. AS is a Lister Institute Research Prize Fellow. None of the authors of this manuscript have a financial interest related to this work.

## Additional information

### Funding

| Funder | Grant reference | Author |
|---|---|---|
| Marie Curie Cancer Care | Core Funding/Programme Grant | Binyam Mogessie, Daniel Roth, Anne Straube |
| Lister Institute of Preventive Medicine | Research Prize | Daniel Roth, Anne Straube |

The funders had no role in study design, data collection and interpretation, or the decision to submit the work for publication.

### Author contributions

BM, Acquisition of data, Analysis and interpretation of data, Drafting or revising the article; DR, ZR, Acquisition of data, Analysis and interpretation of data; AS, Conception and design, Acquisition of data, Analysis and interpretation of data, Drafting or revising the article

### Author ORCIDs

Anne Straube, http://orcid.org/0000-0003-2067-9041

## Additional files

### Supplementary files

- Supplementary file 1. RNAseq_Myoblast_Myocyte. RNA sequencing data for C2C12 myoblasts and 60 hr differentiated myocytes (*Trapnell et al., 2010*) extracted for four regions in MAP4 as shown in *Figure 2—figure supplement 1*.

- Supplementary file 2. Affymetrix Exon Arrays. Affymetrix exon array data (*Pohl et al., 2009*) for four regions in MAP4 as shown in *Figure 2—figure supplement 1*.

- Source code 1. MATLAB code MTdirectionality. Custom MATLAB function to compute directionality of MT growth relative to the main cell axis, generate figure with tracks colour-coded for direction, plot angular histograms and calculate Kuiper statistics relative to a random distribution.

### Major datasets

The following previously published datasets were used:

| Author(s) | Year | Dataset title | Dataset ID and/or URL | Database, license, and accessibility information |
|---|---|---|---|---|
| Pohl A, Smith K, Fujita P, Cline M, Sugnet C | 2009 | Affy Exon Tissues | http://genome-euro.ucsc.edu/cgi-bin/hgTrackUi?hgsid=208289381_jS9DPWNm4JLAAK3Cg8Wwjs2MvakG&g=affyExonTissues | Publicly available as tracks in UCSC Genome Bioinformatics. |

| Author(s) | Year | Dataset title | Dataset ID and/or URL | Database, license, and accessibility information |
| --- | --- | --- | --- | --- |
| Williams B, Marinov G, Trout D, Schaeffer L, Kwan G, Fisher K, De Salvo G, Mortazavi A, Amrhein H, King B | 2012 | Caltech RNA-seq Track | http://genome.ucsc.edu/cgi-bin/hgTrackUi?g=wgEncodeCaltechRnaSeq&hgsid=419817771_YnYIEb0JfLUWZhhJlY3MSCt32Ok9 | Publicly available as tracks in UCSC Genome Bioinformatics. |

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
