## [Decision Letter]

Thank you for sending your work entitled “A novel isoform of MAP4 organises the antiparallel microtubule array required for muscle cell differentiation” for consideration at *eLife*. Your article has been favorably evaluated by Vivek Malhotra (Senior editor) and three reviewers, one of whom is a member of our Board of Reviewing Editors.

The Reviewing editor and the other reviewers discussed their comments before we reached this decision, and the Reviewing editor has assembled the following comments to help you prepare a revised submission.

1) The in vitro experiments require additional controls and quantification:

1A) There is a huge difference in electrophoretic mobility between the endogenous oMAP4 and the recombinant protein used for reconstitution experiments. The validity of the in vitro reconstitution relies completely on the quality of the protein used, so this point requires clarification.

1B) A negative and a positive control for the formation of microtubule bundles should be shown. The negative control would be a protein that does not create microtubule bundles (for example, kinesin-1 motor domain, EB1 or another microtubule-binding protein which does not function as a microtubule-bundling factor in cells). This would be used to exclude artifactual sources of bundling in the assay (e.g., protein misfolding). The positive control would be an established microtubule bundler, such as PRC1. A comparison with an established anti-parallel bundler is important because the provided data do not conclusively prove that oMAP4 a “true” antiparallel bundler (a protein, whose structure dictates a preference for binding to antiparallel microtubule overlaps). The alternative is that oMAP4 creates randomly-oriented bundles with microtubules pointing in both directions. Such bundles would also resist gliding by kinesin motors and splay apart, as shown in Figure 7. To create a random bundle, oMAP4 would need to zipper both parallel and antiparallel overlaps. Figure 6 shows that zippering occurs with higher frequency when the microtubules grow towards each other at ∼180 degrees. This higher frequency is the only evidence that oMAP4 has a structural preference for antiparallel overlaps. But there might be explanations other than structural preference: for example microtubules may be longer when they encounter each other at ∼180 degrees, and longer microtubules zipper more easily. In this context a positive control would be very useful. What does Figure 6 look like for PRC1, for example? This is an essential point, because random bundles and structured, antiparallel bundles have different implications for the structure of oMAP4, for its cellular function and the model in Figure 8. The microtubule-binding domain of MAP4 alone could be another optional control.

1C) The data shown in Figure 5 (loops and bundles in vitro) should be quantified: for example, the length of bundled microtubules, the frequency of looping, or some other objective measure could be determined and compared to a positive and negative control.

2) Experiments in cells require some additional analyses and explanation:

2A) Images and quantification of microtubule organization in cells depleted of mMAP4 and uMAP4, as well as in rescue experiments with the oMAP4 isoform should be included in Figure 3 (now these controls and rescue are only shown for the fusion phenotype, but it is not even clear how exactly the fusion phenotype relates to the microtubule organization). Further, the quality of microtubule images in Figure 3 is less good than in Figure 1, so the microtubule tracing is less convincing. It would be good to improve this, especially if the authors have access to structured illumination microscopy. Some quantification of the extent of microtubule bundling in different RNAi conditions, if possible, would also be useful.

2B) The authors should provide a better description and some explanation of the differences in the expression of embryonic myosin and relocalization of PCM-1. How can these phenotypes be caused by microtubule disorganization? A better discussion would seem appropriate. Likewise, it is not clear how the reduction of the expression in embryonic myosin follows from the provided images, and how this might be connected to the geometry of microtubule arrays. Again, some additional discussion would help. It would also be helpful if the authors explained what “myogenin” is so that a reader does not have to look it up.

3) Other comments:

3A) The expression of oMAP4 in real muscles should be addressed at least by review of database data, because gene expression in a particular cell line does not necessarily reflect the in vivo situation.

3B) In the Discussion, the authors write: “Thus the mechanisms we reveal here for motor and MAP cooperation in the formation of antiparallel microtubule networks are likely to be of importance beyond muscle cells” but do not expound further. This point gets at the potential impact of this work and its appropriateness for *eLife*, a journal meant to appeal to broad audiences. The appeal could be broadened by describing exactly how these mechanisms are important beyond muscle cells, comparing/contrasting oMAP4 with other bundling proteins, and possibly by phrasing the paper in terms of general mechanisms of bundling. Here, the controls would help, because using a positive control will lead naturally to a comparison that places the work in a broader context.

3C) Figure 8 seems to belong in the Results and not in the Discussion. It would also be good if the authors provided some explanation of the fact that the double DHC+oMAP4 depletion has a weaker phenotype that the single ones.

3D) The authors should consider presenting some data as radial histograms, specifically Figures 1 and 3. The point of the data will be clearer.

3E) An explanation for the “o” in oMAP4 would be helpful. Is the “o” for overlap?

3F) The authors give a *p*-value of something like 10^-52. Saying *p*<<0.001 is the norm.

3G) There are no supplemental movies for the EB experiments (e.g., Figure 3). These should be included.

[Editors' note: further revisions were requested prior to acceptance, as described below.]

Thank you for resubmitting your work entitled “A novel isoform of MAP4 organises the antiparallel microtubule array required for muscle cell differentiation” for further consideration at *eLife*. Your revised article has been favorably evaluated by Vivek Malhotra (Senior editor) and a member of the Board of Reviewing Editors. The manuscript has been improved but there are two remaining issues that need to be addressed before acceptance, as outlined below:

1) The quality of the images in Figure 3 is rather poor. Since these data are crucial for your conclusions, we would like to ask you to include additional images and tracings for all conditions shown in Figure 3—figure supplement 1 and Figure 3—figure supplement 2, so that the readers can judge for themselves whether they find the data reliable.

2) The in vitro evidence on oMAP4 as an anti-parallel bundler is still not convincing. For the proposed function, it actually would make sense that oMAP4 can make both parallel and anti-parallel bundles. We would like to suggest you reformulate the Title and downplay the message about anti-parallel bundling in the Abstract.

---

## [Author Response]

*1) The in vitro experiments require additional controls and quantification*:

*1A) There is a huge difference in electrophoretic mobility between the endogenous oMAP4 and the recombinant protein used for reconstitution experiments. The validity of the in vitro reconstitution relies completely on the quality of the protein used, so this point requires clarification*.

Our recombinant protein runs at the correct size for its molecular weight when expressed and purified from *E.coli*. However, expressing the same protein in HeLa cells, causes an increase in apparent molecular weight and the native protein detected by specific antibodies in differentiating muscles cells runs even higher. As proteins are not subject to modifications when expressed recombinantly in bacteria and MAP4 could be subject to different modifications when expressed exogenously in HeLa cells, the observed differences are likely caused by posttranslational modifications. MAP4 is known to be subject to phosphorylation (Semenova et al., Mol Biol Cell 2014) and all MAP4 isoforms as well as other structural MAPs run well above their molecular weight on SDS-PAGE: uMAP4 (West et al., J Biol Chem 1991), mMAP4 (Mangan et al., Development 1996) and other structural MAPs (Noble et al., J Cell Biol 1989; Lee et al., Science 1988; Irminger-Finger et al., J Cell Biol. 1990). We have confirmed the identity of the purified protein using mass spectroscopy, peptides found covered 77% of the proteins with no substantial gaps. We are therefore confident that the protein expressed from our RNAi-resistant rescue constructs mediating oMAP4 function and the purified protein used for in vitro experiments are functionally equivalent.

*1B) A negative and a positive control for the formation of microtubule bundles should be shown. The negative control would be a protein that does not create microtubule bundles (for example, kinesin-1 motor domain, EB1 or another microtubule-binding protein which does not function as a microtubule-bundling factor in cells). This would be used to exclude artifactual sources of bundling in the assay (e.g., protein misfolding). The positive control would be an established microtubule bundler, such as PRC1. A comparison with an established anti-parallel bundler is important because the provided data do not conclusively prove that oMAP4 a “true” antiparallel bundler (a protein, whose structure dictates a preference for binding to antiparallel microtubule overlaps). The alternative is that oMAP4 creates randomly-oriented bundles with microtubules pointing in both directions. Such bundles would also resist gliding by kinesin motors and splay apart, as shown in*
Figure 7*. To create a random bundle, oMAP4 would need to zipper both parallel and antiparallel overlaps.*
Figure 6
*shows that zippering occurs with higher frequency when the microtubules grow towards each other at ∼180 degrees. This higher frequency is the only evidence that oMAP4 has a structural preference for antiparallel overlaps. But there might be explanations other than structural preference: for example microtubules may be longer when they encounter each other at ∼180 degrees, and longer microtubules zipper more easily. In this context a positive control would be very useful. What does*
Figure 6
*look like for PRC1, for example? This is an essential point, because random bundles and structured, antiparallel bundles have different implications for the structure of oMAP4, for its cellular function and the model in*
Figure 8*. The microtubule-binding domain of MAP4 alone could be another optional control*.

As requested we repeated both bundling free in solution and zippering experiments using EB1-GFP as a negative control and PRC1 as a positive control. The new data are shown in Figure 5 and Figure 6. We found that PRC1 was a more efficient bundling protein for free microtubules, but failed to zipper surface attached microtubules. Therefore, we could not compare the antiparallel bias in zippering for PRC1 and oMAP4. Instead, we have followed the additional suggestion to determine the length dependence of zippering and did not find a significant difference between the length distributions of microtubules that are zippered by oMAP4, fail to be zippered, are of parallel or antiparallel orientation. These data are included in the new Figure 6—figure supplement 1. Thus our data show that oMAP4 more efficiently zippers antiparallel microtubules, but can also zipper parallel microtubules. We do think that zippering in either orientation supports an ordered paraxial array and there is no requirement for a preference for its cellular function.

*1C) The data shown in*
Figure 5
*(loops and bundles in vitro) should be quantified: for example, the length of bundled microtubules, the frequency of looping, or some other objective measure could be determined and compared to a positive and negative control*.

As mentioned in response to 1B above, we have repeated these experiments, included PRC1 and EB1 as controls and quantified the occurrence of microtubule bundles formed in solution. The new data are shown in Figure 5.

*2) Experiments in cells require some additional analyses and explanation*:

*2A) Images and quantification of microtubule organization in cells depleted of mMAP4 and uMAP4, as well as in rescue experiments with the oMAP4 isoform should be included in*
Figure 3
*(now these controls and rescue are only shown for the fusion phenotype, but it is not even clear how exactly the fusion phenotype relates to the microtubule organization). Further, the quality of microtubule images in*
Figure 3
*is less good than in*
Figure 1*, so the microtubule tracing is less convincing. It would be good to improve this, especially if the authors have access to structured illumination microscopy. Some quantification of the extent of microtubule bundling in different RNAi conditions, if possible, would also be useful*.

We have performed the requested additional experiments and determined microtubule orientation for uMAP4 and mMAP4 depletion as well as for the oMAP4 rescue experiment. The new data are shown in Figure 3—figure supplement 1. Unfortunately we have no regular access to superresolution microscopy and performed the new experiments identically to the previously existing data from spinning disk images. While it is not possible to trace every single microtubule, we believe the assay is valid and robust as it is reproducible even if different individuals blind to the expected result manually segment microtubules. The density of microtubules in differentiating cells precludes a meaningful analysis of bundling with light microscopy methods.

*2B) The authors should provide a better description and some explanation of the differences in the expression of embryonic myosin and relocalization of PCM-1. How can these phenotypes be caused by microtubule disorganization? A better discussion would seem appropriate. Likewise, it is not clear how the reduction of the expression in embryonic myosin follows from the provided images, and how this might be connected to the geometry of microtubule arrays. Again, some additional discussion would help. It would also be helpful if the authors explained what “myogenin” is so that a reader does not have to look it up*.

We have added a paragraph to the Discussion (first paragraph) that elaborates on this issue, but we cannot do much more than speculate at this point as substantially more research in this area is required to understand the relationship between microtubule organisation, chemical modifications and the myogenic programme. We have added a note to the figure legend explaining that myogenin is used here as a marker for differentiating myoblasts.

*3) Other comments*:

*3A) The expression of oMAP4 in real muscles should be addressed at least by review of database data, because gene expression in a particular cell line does not necessarily reflect the in vivo situation*.

We added a new Figure 2—figure supplement 1 showing RNA sequencing and tissue microarray data that (1) verify our data regarding differential expressing in C2C12 myoblasts and differentiated myocytes and (2) show the expression profile of all MAP4 isoforms in different tissues. oMAP4 is highly expressed in brain and skeletal muscle, while mMAP4 is most prevalent in skeletal muscle and heart. The ubiquitous uMAP4 shows little differential expression.

*3B) In the Discussion, the authors write: “Thus the mechanisms we reveal here for motor and MAP cooperation in the formation of antiparallel microtubule networks are likely to be of importance beyond muscle cells” but do not expound further. This point gets at the potential impact of this work and its appropriateness for* eLife*, a journal meant to appeal to broad audiences. The appeal could be broadened by describing exactly how these mechanisms are important beyond muscle cells, comparing/contrasting oMAP4 with other bundling proteins, and possibly by phrasing the paper in terms of general mechanisms of bundling. Here, the controls would help, because using a positive control will lead naturally to a comparison that places the work in a broader context*.

The additional data showing expression of oMAP4 in brain suggest an importance for oMAP4 in organising the non-centrosomal microtubules in axon and dendrites that remains to be confirmed. MAP4 has so far been considered as the non-neuronal tau equivalent and its role in the presence of the other members of the structural MAP family remain to be elucidated. Further, our additional in vitro experiment show differences in the properties of PRC1 and oMAP4, prompting a number of open questions to understand the structural basis of the ability to zipper microtubules and the directional bias in crosslinking by a largely unstructured protein. We now explain this is the Discussion section.

*3C)*
Figure 8
*seems to belong in the Results and not in the Discussion. It would also be good if the authors provided some explanation of the fact that the double DHC+oMAP4 depletion has a weaker phenotype that the single ones*.

Our model in Figure 8 and described in the Discussion offers a possible explanation why co-depletion of dynein alleviates microtubule alignment. As in many biological processes, imbalance probably causes the biggest problems and the removal of dynein that is responsible for looping movements and the crosslinker oMAP4 that prevents this, probably allows room for other proteins with crosslinking activity such as kinesin-1 and other organising activities such as guided microtubule assembly to dominate. We now moved the data from Figure 8 into Figure 4—figure supplement 1 and mention them in the Results.

*3D) The authors should consider presenting some data as radial histograms, specifically*
Figures 1 and 3*. The point of the data will be clearer*.

As requested we have included radial histograms for the data shown in Figure 1 (Figure 1—figure supplement 1), the new data obtained for uMAP4, mMAP4, oMAP4 depletion and oMAP4 rescue in Figure 3—figure supplement 1, the data shown in Figure 3 (Figure 3—figure supplement 2) and the data shown in Figure 3 and previously shown in Figure 8 (Figure 4—figure supplement 1).

3E) An explanation for the “o” in oMAP4 would be helpful. Is the “o” for overlap?

We of course named the third MAP4 isoform before we knew what it was doing and the true meaning is “other”. However, we agree that this is not satisfactory and as oMAP4 is a microtubule organiser we have decided oMAP4 to stand for organising microtubule-associated protein 4. We included a statement to this effect in the Discussion.

*3F) The authors give a* p*-value of something like 10^-52. Saying* p*<<0.001 is the norm*.

We have amended this as suggested.

*3G) There are no supplemental movies for the EB experiments (e.g.,*
Figure 3*). These should be included*.

We have included the movies of the cells for which tracks are shown in Figure 3 as new Videos 8 and 9.

[Editors' note: further revisions were requested prior to acceptance, as described below.]

*1) The quality of the images in*
Figure 3
*is rather poor. Since these data are crucial for your conclusions, we would like to ask you to include additional images and tracings for all conditions shown in*
Figure 3—figure supplement 1 and Figure 3—figure supplement 2*, so that the readers can judge for themselves whether they find the data reliable*.

We added two examples with traces for each of the 5 conditions shown in Figure 3—figure supplement 1 as suggested. Figure 3—figure supplement 2 contains EB tracks for which an example is shown in Figure 3 and supplementary movies are provided. For the MT orientation data shown in Figure 3 and Figure 4—figure supplement 1, we now provide deconvolved images and movies and hope the quality is sufficiently improved to believe the accompanying traces. Note there are now three examples with traces for control and oMAP4 RNAi data, one each in Figure 3 and two each in Figure 3—figure supplement 1.

*2) The in vitro evidence on oMAP4 as an anti-parallel bundler is still not convincing. For the proposed function, it actually would make sense that oMAP4 can make both parallel and anti-parallel bundles. We would like to suggest you reformulate the Title and downplay the message about anti-parallel bundling in the Abstract*.

We exchanged the word antiparallel with paraxial in the Title and Abstract and throughout the manuscript text where it was more appropriate, i.e. referred to MTs in cells as this describes the arrangement more accurately. oMAP4 shows a bias for zippering antiparallel versus parallel microtubules. We feel we described this accurately in the manuscript, but there isn’t sufficient space in the Abstract to include all detail, especially as we could only assess kinesin forces on antiparallel bundles.